# MLE-RL: REINFORCEMENT LEARNING FOR SELF-IMPROVEMENT IN MACHINE LEARNING AGENTS

## ABSTRACT

Language models have shown significant promise in complex reasoning and coding tasks. However, coding for machine learning engineering presents unique challenges due to the iterative nature of development, long execution times, and the need for continuous self-improvement. In this paper, we introduce MLE-RL trained with reinforcement learning to address these challenges. Our approach reframes the learning process by breaking down long-horizon trajectories into single-step optimizations. We employ a reinforcement learning strategy that selectively learns from the most informative attempts, optimizing the policy on valuable steps. In addition, to overcome context limitations, our agent uses a scaffold with a memory module to store and recall high-performing past solutions, facilitating cumulative learning. The evaluation on the MLE-Bench demonstrates that our MLE-RL-32B achieves 4.9% improvement over the baseline model in the competition ranking on ML tasks and achieves competitive performance against state-of-the-art open-source models like DeepSeek-R1-0528. MLE-RL is open-sourced at https://anonymous.4open.science/r/MLE-RL-CC61

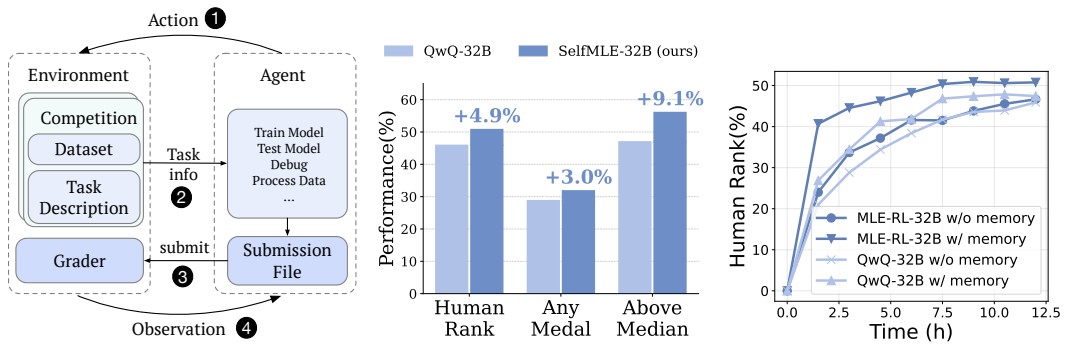

Figure 1: *Left*: Overview of agent-environment interaction in MLE tasks. The agent receives a task description and dataset from the environment, generates and submits solutions, which a grader evaluates to access a performance score for iterative optimization. *Middle*: MLE-RL-32B consistently outperforms baseline methods on three main evaluation metrics(Human Rank, Any Medal and Above Median). *Right*: Both the baseline model(QwQ-32B) and MLE-RL-32B benefit from our memory design, with MLE-RL-32B exhibiting larger gains.

## 1 INTRODUCTION

Language models (LMs) have demonstrated excellent performance on reasoning (Luo et al., 2025; Wang et al., 2024a) and coding (Hui et al., 2024; Zhu et al., 2024) tasks, and tool-augmented LM agents already handle complex tasks, from software engineering (SWE) (Jimenez et al., 2024; Yang et al., 2024) to scientific workflows (Ghafarollahi & Buehler, 2025; Novikov et al., 2025). Unlike traditional single-shot code-generation tasks like coding competitions (Li et al., 2022) or software engineering, machine learning engineering (MLE) focuses on improving system performance over extended periods with limited time budgets (Chan et al., 2025) but no restriction on attempt times. Figure 1(left) illustrates how an LLM handles MLE tasks as an agentic loop: plan the pipeline,

write/run code, inspect results, then iterate, tuning features and hyperparameters, swapping models, fixing errors, using the score as continuous feedback, while retaining the best artifacts and stopping when the budget is exhausted. Since each submission requires code execution, which may take hours, a 12 or 24-hour time window is typically necessary to support adequate iterative experimentation. For example, MLE-Bench (Chan et al., 2025) evaluates LLM's on their best performance on ML tasks within 24 hours. This creates challenges in accurately attributing the sources of improvement, managing heterogeneous reward scales, and leveraging prior work to optimize future experiments. These properties demand LLMs with self-improvement – the ability to accumulate experience, retain/adapt prior solutions, and refine strategies across iterations.

Previous works generally focus on scaling up test-time compute with workflow designs (Liu et al., 2025; Nam et al., 2025), yet few works have paid attention to how to optimize self-improvement abilities of LLMs through training. An MLE task provides an associated dataset, a public test set and a leader board, making it more feasible to evaluate the quality of a solution from an LLM through direct code execution than to obtain proprietary, state-of-the-art solutions. For such easy-to-verify tasks, reinforcement learning has emerged as a powerful and effective strategy. (DeepSeek-AI, 2025; AlphaProof & teams, 2024)

However, the model must iteratively refine its solutions to attain higher performance, introducing additional challenges. 1) Unlike existing reasoning tasks, e.g., math, optimized for single-shot correctness, MLE seeks the best solution within a time budget and tolerates failures. Prioritizing informative and best-performing attempts over training on all attempts is more crucial. 2) Credit assignment is an inherent problem for a multi-step improvement process. 3) While continuous improvement over past experiences is expected, the limited context length of LLMs restricts access to past experiences in multi-turn scenarios.

**Contributions.** To overcome the challenges, we propose MLE-RL, a reinforcement learning framework to foster continuous self-improvement in machine learning engineering (MLE) tasks. MLE-RL trains LLM to learn from valuable past experiences and operates within an agentic scaffold equipped with a memory module. Our contributions are as follows:

First, we propose a reinforcement learning strategy that learns from informative attempts rather than all attempts. To address the credit assignment problem inherent in multi-step interactions, we reframe the task by splitting long-horizon trajectories into single-step optimization units. This enables a more precise attribution of rewards and allows us to apply a curated data selection strategy, optimizing the policy on only the most valuable and informative steps. This entire process is embedded within an asynchronous training framework, enabling efficient and robust policy learning considering the overlong execution time and latency.

Second, to overcome context length limitations and enable the agent to learn from past successes, we introduce a memory module. This module stores high-performing solutions from the agent's history. By randomly selecting a past solution to inform its next attempt, the agent can build upon previous successful experiences that would otherwise be lost, allowing for knowledge accumulation and iterative improvement of its best solutions.

We evaluate MLE-RL on MLE-Bench (Chan et al., 2025), a comprehensive and challenging benchmark for ML agents. As shown in Figure 1, MLE-RL-32B can significantly achieve 4.9% improvement over the baseline model in competition ranking and 9.1% in above median on ML tasks, demonstrating competitive performance to state-of-the-art open-source models. MLE-RL-32B also shows consistently better results at different timestamps in the evaluation stage.

## 2 RELATED WORK

**Code Agents for LLMs.** In recent years, the applications of code AI agents have attracted increasing attention (Holt et al., 2024; Yang et al., 2024; Zhang et al., 2024a). For instance, LLM-based code agents have been widely explored for software engineering (SWE) tasks, where systems such as SWE-agent (Yang et al., 2024), AutoCodeRover (Zhang et al., 2024b), and OpenHands (Wang et al., 2024b) provide frameworks that enable models to autonomously edit code and resolve issues (Jimenez et al., 2024). Beyond agent scaffolds, increasing efforts have focused on improving agent performance on SWE tasks through model training (Pan et al., 2024; Xie et al., 2025) or scale RL-based LLM reasoning for real-world software engineering (Wei et al., 2025).

Machine learning engineering (MLE) has become an emerging domain for evaluating code agents. Framework-driven methods including AIDE (Jiang et al., 2025), ML-Master (Liu et al., 2025), AutoMind (Ou et al., 2025), and MLE-STAR (Nam et al., 2025)employ tree-structured exploration, while scaffolds such as MLAB (Huang et al., 2023) and OpenHands (Wang et al., 2024b) provide general tool-use interfaces for automating ML tasks. Agentic loop systems further incorporate iterative refinement through role separation (Yang et al., 2025). However, most existing efforts are based on comprehensive prompting and scaffold design rather than end-to-end trainable agents. Consequently, how to improve AI agents' performance on MLE tasks through direct training still remains underexplored.

**Reinforcement Learning for Language Models.** Reinforcement learning (RL) has recently become a central approach for enhancing reasoning abilities in large language models, demonstrating substantial gains to mathematical and coding tasks (DeepSeek-AI, 2025; Qwen, 2025; Hou et al., 2025). Typical training paradigms treat generated attempts as approximately i.i.d. samples (DeepSeek-AI, 2025; Qwen, 2025), rely on verifiable answers or reward models to provide supervision (Hou et al., 2025), and apply batch or group-level reward normalization to stabilize optimization (Shao et al., 2024). However, they fail to apply directly to ML tasks due to the non-i.i.d. nature of interactions and the distinctiveness of the reward signal.

## 3 PRELIMINARY

**Iterative Self-improvement for Machine Learning Tasks (MLE)** Following MLE-Bench, the input consists of a machine learning task description and a competition dataset $\mathcal{D}$. The agent generates a solution $s \in \mathcal{S}$, where $S$ represents the solution space and the execution result yield a performance score $h(s) \in \mathcal{R}$ (e.g., accuracy or loss) to reflect the solution's effectiveness. The goal is to find the optimal solution $s^* = \arg\max_{s \in S} h(s)$ within a given inference cost or time limit. To achieve this objective, the search can be cast as direct code generation or as iterative, solution-level self-improvement to make full use of the inference budget.

Formally, given a task, the policy $\pi_\theta$ receives a prompt $x$ and generates an initial solution $s_0$. The model then enters an iterative self-improvement process, where at each step $k$, it updates the current solution $s_k$ to a refined version $s_{k+1}$. During this process, $\pi_\theta$ conditions its generation on the current solution $s_k$ and its feedback $o_k$(e.g., execution traces or evaluation metrics), as well as historical information that may include all or a subset of previous solutions $s_{<k}$ and their feedback signals $o_{<k}$. We define the state at step $k$ as $\tau_k = (s_k, s_{<k}, o_k, o_{<k})$, and the next solution is sampled as: $s_{k+1} \sim \pi_\theta(\cdot \mid \tau_k, x)$. After a predefined time budget, the best solution is selected according to its performance score $h(s)$ on a held-out validation set.

**Reinforcement Learning for LLMs.** Reinforcement learning has been serving a critical role in advancing the reasoning and agent capabilities of LLMs. This paradigm allows LLMs to learn from self-exploration and optimize based on reward signals. In a typical RL process, the policy model $\pi_\theta$ generates a set of $K$ responses, $(\boldsymbol{y}_1, \ldots, \boldsymbol{y}_K)$, for a given input $\boldsymbol{x}$. Each response $\boldsymbol{y}_i$ is then assigned a scalar reward $r(\boldsymbol{x}, \boldsymbol{y}_i)$. The model $\pi_\theta$ is subsequently updated to maximize the expected reward, commonly via an objective function incorporating an advantage term:

$$\mathbb{E}_{\boldsymbol{x} \sim p_{\text{data}}, \boldsymbol{y} \sim \pi_\theta} \frac{1}{K} \sum_i^K A(\boldsymbol{x}, \boldsymbol{y}_i) \log \pi_\theta(\boldsymbol{y}_i | \boldsymbol{x}) \tag{1}$$

Here, $A(\cdot)$ represents the advantage function, often formulated as $A(\boldsymbol{x}, \boldsymbol{y}_i) = \beta(r(\boldsymbol{x}, \boldsymbol{y}_i) - b)$, where $b$ is a crucial baseline that normalizes the reward signal. Group Relative Policy Optimization (GRPO)(Shao et al., 2024) is widely adopted to optimize LLM with RL. For a query $x$ generating a group of responses $\{y_i\}_{i=1}^G$, GRPO defines the advantage $\widehat{A}_i$ for each response $y_i$ as:

$$\widehat{A}_i = \frac{r(x, y_i) - \text{mean}\left(\{r(x, y_i)\}_{i=1}^G\right)}{\text{std}\left(\{r(x, y_i)\}_{i=1}^G\right)}. \tag{2}$$

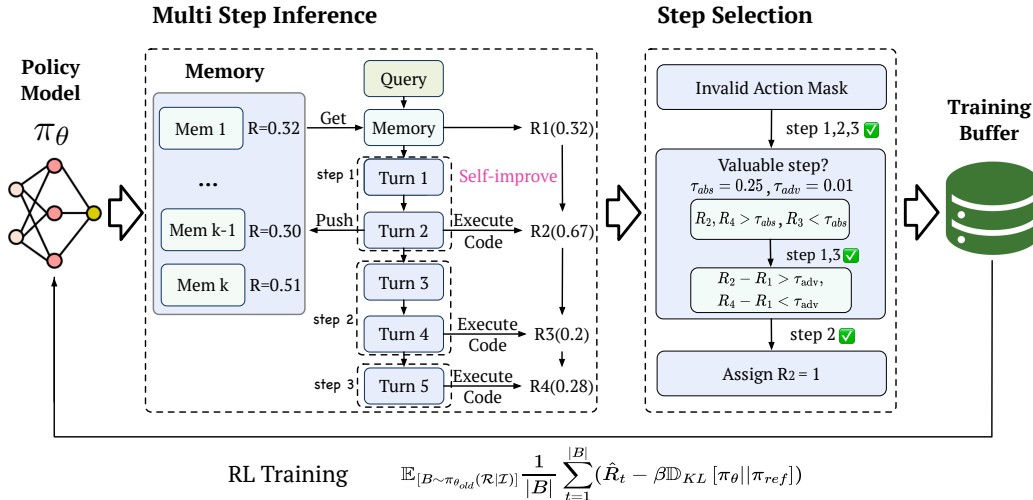

$$\text{RL Training} \quad \mathbb{E}_{[B \sim \pi_{\theta_{old}}(\mathcal{R}|\mathcal{I})]} \frac{1}{|B|} \sum_{t=1}^{|B|} (\hat{R}_t - \beta \mathbb{D}_{KL}[\pi_\theta || \pi_{ref}])$$

Figure 2: Overview of the MLE-RL framework. The policy model interacts with an agentic scaffold equipped with a memory module, which stores and reuses high-quality historical attempts. Data selection based on invalid action masking and valuable step selection retains informative samples, which are collected into a training buffer for policy optimization.

# 4 MLE-RL: RL FOR MACHINE LEARNING ENGINEERING

In this section, we present MLE-RL to advance the self-improvement capabilities of LLMs to solve machine learning engineering (MLE) tasks. The core idea of MLE-RL is to promote the exploration and effective use of past experiences in search and learn from informative and valuable attempts.

To achieve this, we first develop strategies to improve LLMs via reinforcement learning. The idea is to train the LLM to learn from informative attempts rather than the amount of low-value samples. Second, we design an agentic scaffold equipped with a memory module which stores excellent past experiences. The memory enables exploration for improvement based on best practices up to each step. The overview of MLE-RL is illustrated in Figure 2.

## 4.1 OPTIMIZING MLE WITH REINFORCEMENT LEARNING

In this part, we describe how to improve via reinforcement learning (RL). We first reframe the multi-step self-improvement problem as a single-step optimization. For optimization, instead of training on all generated data in RL, we optimize the LLM to learn from the most informative steps of high rewards with curated data selection and reward designs.

**Multi-step self-improvement as a single-step optimization.** Following the design in MLE-Dojo (Qiang et al., 2025), we tackle the ML task as an agent−environment interaction. At time $t$, the agent selects action $a_t \in A$ and receives observation $o_t \in O$. The observation can be the information of the problem, execution results of the code, or the evaluation metric of a machine learning problem. We meticulously select important primitives from the predefined action spaces: `request_info`, `validate_code`, and `execute_code`, as described in appendix B. The agent operates in a multi-turn loop, alternately proposing code or information requests and consuming execution feedback and metric scores. As shown in Table 1, this multi-turn interaction yields consistent gains over the w/o agent baseline, which allows only one submission per trace and thus precludes self-improvement.

With this multi-turn agent scaffold, multiple submissions can be made within a single trace, and each submission is associated with a distinct score. This presents a challenge for credit assignment, as evaluating the entire trace as a single unit makes it difficult to isolate the contribution of the actions that led to a specific submission. To enable a more precise attribution of reward for each successful attempt, we split a multi-turn trajectory into multiple interaction units for training.

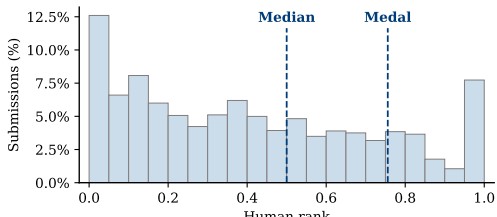 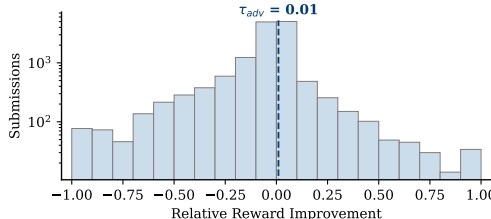

Figure 3: (a) Human rank distribution across all submissions. Median and Medal denotes the human rank median(50%) and the average medal-winning threshold across competitions. A considerable proportion falls below these lines. (b) Distribution of relative reward improvement. More than half of the submissions fall below the threshold $\tau_{adv} = 0.01$, indicating frequent negative optimizations.

Specifically, Each multi-turn interaction can be denoted as a sequence:

$$S = \{x, a_1, o_1, ..., a_N, o_N\} \qquad (3)$$

where $x$ denotes the input problem, $a_i$ the assistant's action at turn $i$, and $o_i$ its environment feedback. Corresponding to the description of action space in Section 4.1, we view actions between two *execute_code* actions that produce a valid submission as a *step*.

For each training instance at turn $k$ ($1 \leq k \leq N$), the model is provided with the entire history up to the current turn. The resulting training sample is constructed as:

$$S_k = \left( x, (a_1^1, o_1^1, \ldots a_1^{N_1}, o_1^{N_1}), \ldots, (a_M^1, o_M^1, \ldots a_M^{N_M}, o_M^{N_M}) \right) \qquad (4)$$

where $(a_i^1, o_i^1, \ldots a_i^{N_i}, o_i^{N_i})$ denotes the $i$-th step, $a_i^j$ the $j$-th action of step i, $N_i$ the number of actions within step i, and M the number of total steps within the trace. $a_M^{N_M}$ is always *execute_code*.

Each step is considered as a training instance during RL training, with all turns within the step assigned an identical reward so that every contributing action receives equal credit.

**Training with valuable steps.** For MLE tasks, we propose to optimize the model with the most informative steps rather than all generated attempts. The overall target is to train the model to improve over the previous solutions and achieve superior performance. Figure 3 illustrates that a significant portion of solutions falls below the median and medal-winning thresholds, indicating the presence of many suboptimal solutions. Furthermore, more than half of the attempts fail to achieve reasonable improvement over the previous ones. Therefore, effective data selection is essential to prevent suboptimal and negatively optimized instances from destabilizing the learning process.

To improve training data quality and enhance training robustness by prioritizing high-quality solutions and mitigating suboptimal responses, we employ two specialized filters as follows:

- `Mask invalid actions`. We introduce a mask strategy to prevent the model from collapsing into invalid tool-using output. Instead of assigning a negative reward to invalid format output, we mask the loss on agent responses that result in an invalid format (e.g., invoking incorrect tool calls or exceeding length limits) or an invalid submission. This allows the model to leverage the full contextual trajectory for learning without reinforcing erroneous outputs, thereby contributing to preventing the generation of undesirable behaviors.
- `Valuable step selection`. To select valuable steps that are beneficial to training, we only retain the steps that attain a reward exceeding a competition-specific threshold $\tau_{\mathrm{abs}}$. This threshold is carefully calibrated to retain approximately $30\%$ of the highest-scoring data for training in each competition, thus preventing the model from repeatedly drawing on suboptimal solutions. In addition, The model is expected to iteratively refine solutions based on prior experience. However, as depicted in Figure 3 (b), generated solutions constitute numerous negative optimizations. To steer the model to generate improved solutions than the reference ones and prevent negative optimization, we retain the step as training data *only* if the relative reward improvement of the current submission, defined as the reward improvement between the current and the first valid solution generated within that same trace, surpasses a predefined threshold $\tau_{\mathrm{adv}}$. This reinforces substantive improvements over the preceding solutions in a given interaction.

However, as the training goes on with iterative improvements, the average performance would gradually grow up and thus a static data filtering strategy is problematic. To address the non-stationarity of execution outcomes arising from stochasticity and shifting data splits, we adopt a dynamic, competition-specific reward normalization. For each instance with raw reward $r_i = \mathrm{HumanRank}_i$, calculated as $\mathrm{HumanRank}_i = 1 - \frac{p}{N}$, where p is the solutions leaderboard position and N is the total number of human competitors, we compute a running mean over a historical window $\mathcal{H}_i$ containing the $W$ most recent rewards from the same competition, prune outliers more than two standard deviations below the window mean, and obtain the normalized reward:

$$\bar{r}_i = r_i - \frac{1}{|\mathcal{H}'_i|} \sum_{j \in \mathcal{H}'_i} r_j, \quad \mathcal{H}'_i = \{\, j \in \mathcal{H}_i \mid r_j \geq \mu_{\mathcal{H}_i} - 2\sigma_{\mathcal{H}_i} \,\}.$$

Notably, since traces utilizing memory benefit from accumulated past experiences, we maintain separate running mean windows for memory and reset traces to prevent bias against those starting from scratch. During RL training, we actually use $r_i = 1$ if $\bar{r}_i > \tau_{\mathrm{rm}}$ else $r_i = 0$, where $\tau_{\mathrm{rm}}$ is the running mean threshold that adaptively distinguishes meaningful improvements from noise. This implementation amplifies the contribution of positive samples and masks the gradient of all negative samples. This running filter helps the training set emphasizes substantive solution changes and thereby supports robust policy learning under inevitable evaluation noise.

**Asynchronous Training for RL.** To address the challenge of high reward latency originating from code execution, we adopt a fully decoupled training and data generation framework. Specifically, data generation workers are asynchronously executed based on our scaffold, accumulating generated samples into a data pool. Once the number of samples reaches a predefined batch size $B$, the training process is triggered to update the model parameters using the latest batch of data. By decoupling training and data generation, we can flexibly scale up the number of data generators, allowing the throughput of data generation and training to be balanced, and thus mitigating the impact of slow reward feedback on overall training efficiency. For policy optimization, we employ REINFORCE (Williams, 1992) with the reward designs stated above.

## 4.2 AGENTIC SCAFFOLD WITH MEMORY

Multi-step self-improvement in machine learning engineering learns from past iterations to find better solutions, but limited context length of LLM constrains the access to its history of attempts. Existing methods either discard all history when the context window fills, or restrict learning to a subset of prior iterations with predefined workflow, at the cost of agent design flexibility.

To enable the model to explore based on historical experiences and mitigate context limitations, we introduce a memory module that stores valuable historical attempts. The module maintains a pool of high-scoring solutions together with metadata (score, trace context, trace identifier) and affords two operations: `push` and `get`:

- `push`: If a new solution $s$ discovered by the policy model exceeds the memory pool's minimum score, insert $s$ into the memory pool or replace the lowest-scoring entry.

- `get`: Randomly sample a solution from the pool to condition the next trajectory.

When a new trace begins due to context limits, the agent calls `get` to warm-start from a good prior solution, refining rather than restarting from scratch. To preserve diversity and prevent mode collapse, we compute abstract syntax tree (AST) similarity between candidates and pool members, retaining only the highest-scoring solution among those whose similarity exceeds a threshold $T_{\mathrm{ast}}$. To encourage exploration, the agent also restarts from scratch with a probability $p$, denoted as *reset ratio*. We denote traces restarting from scratch as *reset traces* and traces warm-starting from a memory solution as *memory traces*. Overall, the memory module balances exploitation of past successes with exploration of new strategies. The agent achieves an unbounded horizon of self-improvement: local iterations proceed within a trace, and global progress persists through memory-driven restarts.

## 5 EXPERIMENTS

### 5.1 SETUP

**Training Details.** Reinforcement learning (RL) training is conducted based on the QwQ-32B model (Qwen, 2025), with a KL coefficient $\beta$ of 0, a learning rate of $1 \times 10^{-6}$, and a training batch size of 64. The maximum context lengths for inputs and responses are set to 65536 and 16384, respectively. The rollout model parameters are synchronized with the latest policy model every 5 training steps. For data generation, we set both temperature and top-$p$ to 1 for sampling diversity. Each generation trace is constrained to a maximum of 15 turns. To manage the search process, we maintain the best-solution pool of 5 candidates for subsequent iterations and set the AST similarity threshold $T_{\text{ast}}$ to 0.9. To foster exploration, we set the reset ratio $p$ to 70%. We set the advantage filter threshold $\tau_{adv}$ to 0.01. During training, the selection of solutions for the memory module is based on their test set scores, even though these scores are not visible to the model during rollout. Conversely, during evaluation, all selections for the memory module exclusively utilize validation set scores to avoid data leakage.

**Training Dataset.** Our training corpus consists of 200 Kaggle competitions, including 97 open source data from MLE-Dojo(Qiang et al., 2025) and 103 tasks privately collected from the official Kaggle competition platform[1].

**Hardware Configurations.** In our experimental setup, the agents execute within Ubuntu 20.04 Docker containers configured with the dataset and Python packages commonly employed in machine learning engineering. Computational resources for rollouts include 128 vCPUs, 700 GB of memory, and NVIDIA A10 GPUs. For policy training, we utilize NVIDIA H800 GPUs for a total wall-clock training time of 20 hours.

**Evaluation setting.** To assess model effectiveness on machine learning tasks, we perform a standardized evaluation on two benchmarks: the full 75-competition MLE-Bench and the 22-competition MLE-bench-Lite subset. The time budgets are 24 hours for the full benchmark and 12 hours for the subset. Our agent scaffold, configured with a 0.5 reset ratio and a memory size of 3, handles automated solution generation and submission. Performance is determined by ranking the results against human competitors on the official Kaggle leaderboards. Each experiment is repeated for three times, and we report the average evaluation metrics, including Human Rank, Any Medal, and related metrics, together with their standard deviation.

### 5.2 EVALUATION RESULTS

Table 1 presents the performance comparison of various models and scaffolds on MLE-bench-Lite respectively. MLE-RL demonstrate consistent performance improvements over the QwQ-32B baseline across all evaluation metrics, including Human Rank(+4.9%), Above Median(+0.1%), and Any Medal(+3.0%) for MLE-Bench-Lite. Specifically, Human Rank measures the percentage of human competitors that the agent outperforms, averaged across all competitions. Any Medal denotes the proportion of competitions where the agent wins at least one medal. Similarly, Table 2 shows the performance on MLE-Bench. It can be observed that MLE-RL still achieves remarkable improvement over the baseline method. These results indicate the effectiveness of our approach in improving task performance on MLE-Bench.

### 5.3 ABLATION STUDY

**Ablation Study on Data Selection Strategies.** Due to the computational intensity and slow convergence of RL, we evaluate the impact of different data selection strategies using Supervised Fine-Tuning(SFT) instead, performing SFT experiments on datasets derived directly from RL rollouts.

As shown in Table 4, model performance consistently improves as more comprehensive data selection strategies are adopted. Training with the full dataset, which includes samples exhibiting format errors and invalid submissions, leads to a decline in performance relative to the QwQ-32B baseline, indicating that exposure to error-prone data can negatively affect the model's ability to generalize

---

[1]https://www.kaggle.com/competitions

Table 1: Experimental results on MLE-Bench-Lite. All baselines are evaluated with the agent scaffold without the memory module. The w/o agent setting operates under our agent scaffold but restricts models to a single submission per trace. All reported metrics are percentages (%).

| Model | Human Rank | Above Median | Bronze | Silver | Gold | Any Medal |
|---|---|---|---|---|---|---|
| gpt-4o-2024-08-06 | $37.5_{\pm3.1}$ | $31.8_{\pm3.7}$ | $3.0_{\pm2.1}$ | $3.0_{\pm2.1}$ | $13.6_{\pm0.0}$ | $19.7_{\pm2.1}$ |
| DeepSeek-v3 | $38.9_{\pm1.3}$ | $36.4_{\pm3.7}$ | $0.0_{\pm0.0}$ | $6.1_{\pm2.1}$ | $13.6_{\pm3.7}$ | $19.7_{\pm2.1}$ |
| DeepSeek-R1-0528 | $49.0_{\pm5.3}$ | $53.0_{\pm5.6}$ | $3.0_{\pm5.0}$ | $6.1_{\pm3.4}$ | $20.5_{\pm4.4}$ | $29.5_{\pm4.4}$ |
| Qwen3-235B-A22B-thinking-2507 | $40.5_{\pm1.1}$ | $40.9_{\pm0.0}$ | $4.6_{\pm3.7}$ | $4.6_{\pm3.7}$ | $15.2_{\pm4.3}$ | $24.2_{\pm2.1}$ |
| Qwen3-32B | $42.4_{\pm4.2}$ | $42.4_{\pm7.7}$ | $0.0_{\pm0.0}$ | $3.0_{\pm2.1}$ | $19.7_{\pm2.1}$ | $22.7_{\pm3.7}$ |
| QwQ-32B (*w/o* agent) | $33.7_{\pm3.1}$ | $30.3_{\pm4.3}$ | $3.0_{\pm2.1}$ | $6.1_{\pm2.1}$ | $13.6_{\pm3.7}$ | $22.7_{\pm0.0}$ |
| QwQ-32B | $45.9_{\pm4.2}$ | $47.0_{\pm4.3}$ | $3.0_{\pm4.3}$ | $4.6_{\pm0.0}$ | $21.2_{\pm7.7}$ | $28.8_{\pm5.7}$ |
| MLE-RL-32B (Ours) | $\mathbf{50.8}_{\pm1.0}$ | $56.1_{\pm2.1}$ | $6.1_{\pm2.1}$ | $3.0_{\pm2.1}$ | $22.7_{\pm3.7}$ | $\mathbf{31.8}_{\pm3.7}$ |

Table 2: Experimental results on MLE-Bench full set. All reported metrics are percentages (%).

| | Valid Submission | Above Median | Bronze | Silver | Gold | Any Medal |
|---|---|---|---|---|---|---|
| **MLAB** (Huang et al., 2023) | | | | | | |
| gpt-4o-2024-08-06 | $44.3_{\pm2.6}$ | $1.9_{\pm0.7}$ | $0.0_{\pm0.0}$ | $0.0_{\pm0.0}$ | $0.8_{\pm0.5}$ | $0.8_{\pm0.5}$ |
| **OpenHands** (Wang et al., 2024b) | | | | | | |
| gpt-4o-2024-08-06 | $52.0_{\pm3.3}$ | $7.1_{\pm1.7}$ | $0.4_{\pm0.4}$ | $1.3_{\pm0.8}$ | $2.7_{\pm1.1}$ | $4.4_{\pm1.4}$ |
| **AIDE** (Jiang et al., 2025) | | | | | | |
| gpt-4o-2024-08-06 | $54.9_{\pm1.0}$ | $14.4_{\pm0.7}$ | $1.6_{\pm0.2}$ | $2.2_{\pm0.3}$ | $5.0_{\pm0.4}$ | $8.7_{\pm0.5}$ |
| o1-preview | $82.8_{\pm1.1}$ | $29.4_{\pm1.3}$ | $3.4_{\pm0.5}$ | $4.1_{\pm0.6}$ | $9.4_{\pm0.8}$ | $16.9_{\pm1.1}$ |
| Deepseek-R1-0528 | $78.6_{\pm0.0}$ | $34.6_{\pm0.0}$ | $2.7_{\pm0.0}$ | $4.0_{\pm0.0}$ | $8.0_{\pm0.0}$ | $14.7_{\pm0.0}$ |
| **Agent Scaffold (Ours)** | | | | | | |
| QwQ-32B | $63.3_{\pm0.7}$ | $22.7_{\pm0.0}$ | $1.3_{\pm0.0}$ | $2.7_{\pm0.0}$ | $8.0_{\pm1.3}$ | $12.0_{\pm1.3}$ |
| MLE-RL-32B (Ours) | $67.3_{\pm2.0}$ | $25.3_{\pm2.7}$ | $3.3_{\pm0.7}$ | $2.0_{\pm0.7}$ | $8.7_{\pm0.7}$ | $\mathbf{14.0}_{\pm0.7}$ |

Table 3: Experimental results of RL-trained (MLE-RL-32B) and self-distilled (MLE-RL-32B-S) models on MLE-Bench-Lite and MLE-Bench. All reported metrics are percentages (%).

| Model | Human Rank | Above Median | Bronze | Silver | Gold | Any Medal |
|---|---|---|---|---|---|---|
| **MLE-Bench-Lite** | | | | | | |
| MLE-RL-32B | $50.8_{\pm1.0}$ | $56.1_{\pm2.1}$ | $6.1_{\pm2.1}$ | $3.0_{\pm2.1}$ | $22.7_{\pm3.7}$ | $31.8_{\pm3.7}$ |
| MLE-RL-32B-S | $\mathbf{51.7}_{\pm2.5}$ | $53.0_{\pm7.7}$ | $7.6_{\pm2.1}$ | $7.6_{\pm4.3}$ | $18.2_{\pm3.7}$ | $\mathbf{33.3}_{\pm4.3}$ |
| **MLE-Bench** | | | | | | |
| MLE-RL-32B | $23.1_{\pm0.8}$ | $25.3_{\pm2.7}$ | $3.3_{\pm0.7}$ | $2.0_{\pm0.7}$ | $8.7_{\pm0.7}$ | $14.0_{\pm0.7}$ |
| MLE-RL-32B-S | $\mathbf{26.9}_{\pm0.8}$ | $26.2_{\pm0.6}$ | $2.2_{\pm1.7}$ | $3.6_{\pm1.7}$ | $10.7_{\pm0.0}$ | $\mathbf{16.5}_{\pm1.7}$ |

and solve ML tasks. Employing invalid action mask results in a clear improvement, confirming that basic filtering to remove invalid submissions is beneficial. Building upon this, value selection yields the best overall performance, suggesting that concentrating training on high-quality samples further enhances model capability.

**Ablation study on RL reward assignment.** In Section 4.1, we utilize a running filter to exclude samples that yield a negative reward after normalization, while assigning a reward of 1 to all positive

Table 4: Ablation study on different data selection strategies. All reported metrics are percentages (%).

| | # Data | Human Rank | Above Median | Any Medal |
|---|---|---|---|---|
| QwQ-32B | - | $45.9_{\pm 4.2}$ | $47.0_{\pm 4.3}$ | $28.8_{\pm 5.7}$ |
| All data | 100% | $43.1_{\pm 0.8}$ | $47.0_{\pm 5.7}$ | $25.8_{\pm 2.1}$ |
| + invalid action mask | 62.1% | $48.5_{\pm 2.3}$ | $54.6_{\pm 3.7}$ | $30.3_{\pm 4.3}$ |
| + value selection | 16.2% | $49.2_{\pm 2.7}$ | $48.5_{\pm 4.3}$ | $31.8_{\pm 3.7}$ |

samples. To analyze the impact of this technique, we perform an ablation study where the original normalized rewards are used directly for the RL training.

Table 5 presents the effects of the competition-specific reward normalization on RL training. Using normalized rewards with both positive and negative values leads to a noticeable decline in performance compared to the fixed-reward formulation. This degradation indicates that while the filtered data already provide sufficiently informative supervision for reinforcement learning, signed rewards introduce additional noise and instability into the optimization process. Moreover, the use of a binary reward addresses the challenge that the distribution of Human Rank scores can vary significantly across different competitions, which could otherwise introduce competition-specific bias.

Table 5: Ablation study on effects of reward assignment for RL training.

| Agent | Human Rank (%) | Above Median(%) | Any Medal (%) |
|---|---|---|---|
| MLE-RL-32B | $50.8_{\pm 2.0}$ | $56.1_{\pm 2.1}$ | $31.8_{\pm 3.2}$ |
| MLE-RL-32B (normalized reward) | $46.3_{\pm 2.1}$ | $52.3_{\pm 4.5}$ | $27.3_{\pm 0.00}$ |

## 5.4 ANALYSIS

**Effects of agent memory.** To assess the memory module's impact on the agent's self-improvement, we evaluated QwQ-32B with a 0.5 reset ratio, ensuring a balanced distribution of memory and reset traces. For a fair comparison, we only analyzed competitions where both trace types produced at least one valid submission.

As shown in Figure 4(left), reset traces quickly plateau after exhausting the benefits of random exploration. In contrast, memory traces show continuous improvement. Although initially delayed while the memory populates with solutions from the reset traces, they leverage these stored solutions to achieve sustained improvement and ultimately outperform their randomly-initialized counterparts. Furthermore, a 12-hour evaluation illustrated in Figure 4(middle) shows that both MLE-RL-32B and QwQ-32B benefit from the incorporation of the memory module, achieving higher Human Rank and Any Medal rates. The more substantial gain in MLE-RL-32B suggests that RL training better equips the model to leverage historical solutions from memory.

**Effects of self-distillation.** We further conducted a self-distillation experiment. We collected rollout data across multiple runs of our RL experiments and build a self-distillation dataset consisting of a large number of high-quality samples from these runs. The resulting dataset can be equally viewed as the product of a single, long-running experiment. We finetune QwQ-32B using the dataset with offline SFT, leading to MLE-RL-32B-S. As is shown in Table 3, the self-distillation shows even better performance over single-run RL across both MLE-Bench-Lite and MLE-Bench. This suggests that data aggregated from multiple RL runs offers a richer and more diverse signal for supervised finetuning, akin to the effect of scaling up inference compute with prolonged RL training.

**Step-wise analysis on Progressive improvement via MLE-RL refinement.** To better evaluate model performance and reduce the impact of code execution time variability across competitions, we analyze model results over valid submission steps. Specifically, for each competition, we consider up to the first k valid submissions (using all available steps if fewer than k) to compute its performance, and then average these scores across all competitions to obtain the overall step-k per-

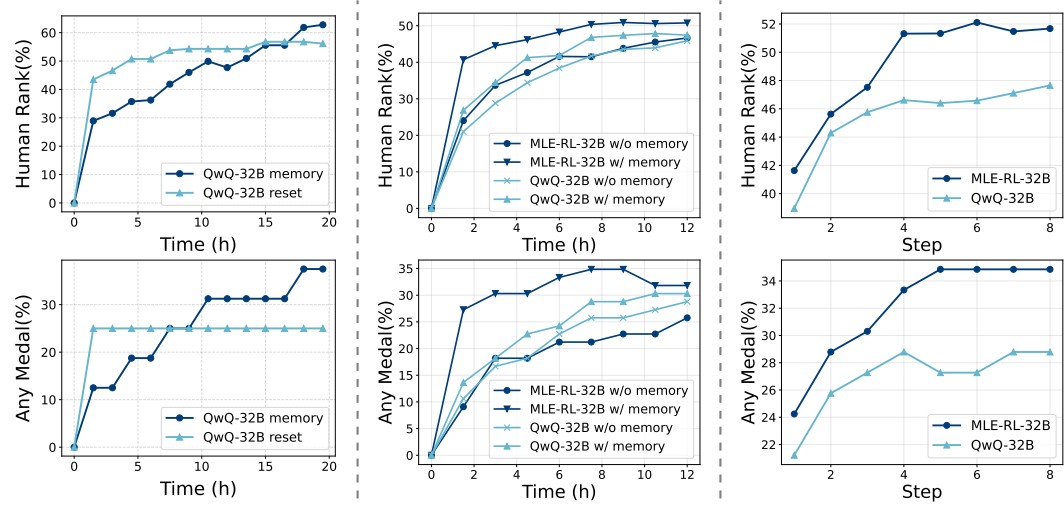

Figure 4: *Left*: Performance comparison of memory trace and reset trace over time within one QwQ-32B evaluation run, with memory traces eventually outperforming reset traces. *Middle*: Performance of QwQ-32B and MLE-RL-32B $w/$ and $w/o$ memory module. Both models benefit from memory design, with MLE-RL-32B exhibiting larger gains. *Right*: Step-wise performance for QwQ-32B and MLE-RL-32B.

formance. Figure 4(right) compares step-wise performance of QwQ-32B and MLE-RL-32B. The improvement of MLE-RL-32B over QwQ-32B at step 1 indicates that training enhances the model's capability to directly generate a valid solution from scratch, while its continued improvement in later steps indicates enhanced iterative refinement capabilities.

## 6    CONCLUSION

This work presents MLE-RL, a LLM agent trained with reinforcement learning(RL) to solve machine learning engineering(MLE) tasks. By reframing long-horizon iterative trajectories into single-step optimizations and selectively learning from informative attempts, our RL strategy achieves consistent improvements in task performance. Furthermore, the integration of a memory module enables agents to retain and reuse high-quality solutions, facilitating sustained self-improvement beyond context length limitations. These findings highlight the effectiveness and potential of our approach for training ML agents to advance autonomous ML research.

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

## A USE OF LLMs

Large language models (LLMs) were used solely for language polishing and grammar refinement during manuscript preparation. All research ideas, methodologies, experiments, and analyses were independently conceived, designed, and validated by the authors.

## B ACTION SPACE

Our work follows the action space specification defined by MLE-Dojo(Qiang et al., 2025). The action space consists of three key primitives:

1. `request_info`: This action grants the agent full access to all competition-related resources required to solve the task. Specifically, it provides access to:
   - **Competition background**: A concise overview of the historical context and the evolving research challenges.
   - **Goal description**: A clear definition of the specific objectives, desired outcomes, and evaluation metrics.
   - **Sample submission**: Templates that exemplify the expected format and content structure for submissions.
   - **Data folder structure**: The organization and naming conventions of the provided datasets, aiding in user access and navigation.

   The information returned is presented in full, without any analysis, filtering, or pruning.

2. `validate_code`: This primitive performs syntax checks and lightweight runtime execution, returning error logs and execution outputs. It is used exclusively for debugging and analysis purposes and does not correspond to an official competition submission.

3. `execute_code`: This primitive executes the submitted code fully, performing submission, verification, and evaluation according to the competition's metric. Each invocation corresponds to a formal competition submission.

## C SCAFFOLD COMPARISON

Table 6 shows the experimental results on MLE-Bench-Lite using four scaffolds: the single-submission per trace setting without self-improvement (w/o agent), the commonly used tree-search framework AIDE, our agent scaffold without the memory module, and MLE-RL-32B. Among these variants, MLE-RL-32B achieves the best overall performance across all metrics, demonstrating the effectiveness of the proposed design.

Table 6: Experimental results on MLE-Bench-Lite of different scaffold. All reported metrics are percentages (%).

| Model | Human Rank | Above Median | Any Medal |
|---|---|---|---|
| QwQ-32B (*w/o* agent) | 33.7 | 30.3 | 22.7 |
| QwQ-32B (AIDE) | 40.3 | 45.5 | 25.0 |
| QwQ-32B (w/o memory) | 45.9 | 47.0 | 28.8 |
| MLE-RL-32B (Ours) | **50.8** | **56.1** | **31.8** |

## D PERFORMANCE ACROSS COMPETITIONS

Table 7 presents a per-competition analysis of results on the MLE-Bench-Lite benchmark, comparing MLE-RL-32B with QwQ-32B across 22 machine learning tasks. Across the 22 evaluated machine learning competitions, MLE-RL-32B demonstrates superior capability by achieving a higher mean Human Rank (HR) in the majority of tasks, indicating its general effectiveness. This is particularly evident in challenges like histopathologic-cancer-detection, where it scored an impressive

| Tasks | QwQ-32B | | MLE-RL-32B | |
|---|---|---|---|---|
| | HR (%) | AM (%) | HR (%) | AM (%) |
| aerial-cactus-identification | $79.1_{\pm 21.4}$ | **33.3** | **$87.7_{\pm 0.0}$** | – |
| aptos2019-blindness-detection | **$73.4_{\pm 1.9}$** | – | $69.5_{\pm 4.6}$ | – |
| denoising-dirty-documents | $41.4_{\pm 13.9}$ | – | **$69.8_{\pm 5.2}$** | **100.0** |
| detecting-insults-in-social-commentary | $99.3_{\pm 0.9}$ | **100.0** | **$100.0_{\pm 0.0}$** | **100.0** |
| dog-breed-identification | $46.3_{\pm 4.1}$ | – | **$47.1_{\pm 3.8}$** | – |
| dogs-vs-cats-redux-kernels-edition | **$100.0_{\pm 0.0}$** | **100.0** | **$100.0_{\pm 0.0}$** | **100.0** |
| histopathologic-cancer-detection | $49.5_{\pm 49.5}$ | 50.0 | **$99.1_{\pm 0.0}$** | **100.0** |
| jigsaw-toxic-comment-classification-challenge | **$29.0_{\pm 6.8}$** | – | $25.7_{\pm 9.3}$ | – |
| leaf-classification | $60.1_{\pm 1.1}$ | – | **$61.0_{\pm 4.7}$** | – |
| mlsp-2013-birds | – | – | $0.0_{\pm 0.0}$ | – |
| new-york-city-taxi-fare-prediction | **$0.1_{\pm 0.0}$** | – | **$0.1_{\pm 0.0}$** | – |
| nomad2018-predict-transparent-conductors | **$58.8_{\pm 2.8}$** | **100.0** | $55.6_{\pm 5.2}$ | 66.7 |
| plant-pathology-2020-fgvc7 | **$98.5_{\pm 1.0}$** | **100.0** | $90.1_{\pm 13.5}$ | 66.7 |
| random-acts-of-pizza | **$57.3_{\pm 20.4}$** | **33.3** | $41.1_{\pm 17.0}$ | – |
| ranzcr-clip-catheter-line-classification | $3.3_{\pm 3.3}$ | – | **$6.4_{\pm 6.4}$** | – |
| siim-isic-melanoma-classification | – | – | $23.8_{\pm 17.0}$ | – |
| spooky-author-identification | **$48.0_{\pm 7.6}$** | – | $44.1_{\pm 18.4}$ | – |
| tabular-playground-series-dec-2021 | **$100.0_{\pm 0.0}$** | **100.0** | **$100.0_{\pm 0.0}$** | **100.0** |
| tabular-playground-series-may-2022 | **$41.4_{\pm 1.2}$** | – | $21.3_{\pm 5.8}$ | – |
| text-normalization-challenge-english-language | – | – | – | – |
| text-normalization-challenge-russian-language | **$13.9_{\pm 0.0}$** | – | $1.2_{\pm 0.0}$ | – |
| the-icml-2013-whale-challenge-right-whale-redux | $70.4_{\pm 29.4}$ | **66.7** | **$76.4_{\pm 24.5}$** | **66.7** |

Table 7: Per-competition results on the 22 tasks from MLE-BENCH-LITE, comparing QwQ-32B with our MLE-RL-32B. Each entry reports the mean Human Rank (HR) and Any Medal (AM) over three runs. Overall, MLE-RL-32B achieves generally higher HR than QwQ-32B.

99.1% HR compared to QwQ-32B's 49.5%, and in denoising-dirty-documents, with a 69.8% HR versus 41.4%. In terms of the Any Medal (AM) metric, MLE-RL-32B also maintains high consistency, frequently reaching or matching top-tier success rates across diverse domains.

