# OpenReview forum: "MLE-RL: Reinforcement Learning for Self-Improvement in Machine Learning Agents"
_ICLR.cc/2026/Conference — Submitted to ICLR 2026_

### Official Review · Reviewer_Gbpo · 2025-10-27

**Soundness:** 4
**Presentation:** 3
**Contribution:** 3
**Rating:** 4
**Confidence:** 4

**Summary:**

MLE-RL addresses the challenge of training LLMs for machine learning engineering (MLE) tasks that require long-horizon, iterative self-improvement through two key contributions. First, it proposes a reinforcement learning strategy that decomposes multi-turn agent trajectories into single-step optimization units (where each "step" represents actions between two valid code submissions) to enable precise credit assignment. Rather than training on all generated attempts, it selectively learns from informative steps using three filters: invalid action masking (removes format errors), valuable step selection (retains approximately top 30%), and dynamic running mean normalization that maintains separate windows for memory-based and scratch trajectories to filter out negative samples. To overcome context length limitations, the MLE-RL introduces a memory module equipped with add and read operations that maintains a pool of high-scoring solutions from past attempts, using AST similarity filtering to maintain diversity. This enables both local (within-trace) and global (cross-trace via memory) self-improvement. MLE-RL-32B demonstrates competitive performance with state-of-the-art models like DeepSeek-R1-0528 despite using only 32B parameters.

**Strengths:**

1. This paper tackle the hard problem of training llms to learn long trajectory tasks ( Kaggle ML tasks was used in this case ). The proposal of trajectory decomposition and rollout data selection is novel.

2. The MLE-bench results are quite competitive given the scale of the model is just 32B. This shows that MLE-RL is effective in training a small models to specialize in highly complex and long trajectory task well.

**Weaknesses:**

This paper lacks the required details to reproduce ( lack of training data transparency, training hyperparams is least minimum and no details of what hardware or budget does it cost ).

Analysis section is bit weak on the side specifically the paper only trains one single model : QwQ-32B, I expect in the analysis or ablation to train MLE-RL on a smaller LLMs to ensure its reproducible even for slightly lower compute budget institutes. The other weakness are addressed in my major concerns section.

Only one dataset is evaluated which means questioning the effectiveness of this method but this is minor since no similar benchmarks are close enough to MLE-bench ( minor concern ).

**Questions:**

**Major concerns:**

1. Which specific training data is the MLE-RL trained on? Throughout the entire paper, it does not mention what is the source of train set, as MLE-bench do not contain any train set at all, all the 75 tasks were classified as test set.

2. Since QwQ is a reasoning model does the reasoning budget has any affect on the final score？

**Minor concerns:**

3. Does the competitive performance from distillation (Sec 5.4) suggests that rejection sampling finetune might be a strong baseline? Since based on the paragraph the distillation just means its trained on the positive rollouts from the RL experiments rather than responses from the final RL model ( which is how Qwen3, Deepseek distillation is done ).

4. What kind of training framework was used in the training of MLE-RL? Is it a proprietary one or existing framework like verl? My understanding is that only MLE-dojo was mentioned but its just a environment to run these rollouts.

5. What hardware is used to trained the MLE-RL, my understanding is the A10 GPU mentioned is just for rollout of the MLE-bench experiments.

6. Typo at line 194: “an agentenvironment interaction”

---

> ### Author Response · Authors · 2025-11-23
>
> We sincerely thank you for your thoughtful and constructive feedback.  We now provide point-by-point responses to elaborate further:
> > **Weakness 1:** This paper lacks the required details to reproduce ( lack of training data transparency, training hyperparams is least minimum and no details of what hardware or budget does it cost ).
>
> > **Question 1:** Which specific training data is the MLE-RL trained on? Throughout the entire paper, it does not mention what is the source of train set, as MLE-bench do not contain any train set at all, all the 75 tasks were classified as test set.
>
> > **Question 5:** What hardware is used to trained the MLE-RL, my understanding is the A10 GPU mentioned is just for rollout of the MLE-bench experiments.
> We provide the following clarifications regarding the training details to ensure reproducibility:
>
> #### **Training Dataset Details**
>
> Our training corpus consists of **200 Kaggle competitions**, including **97 open-source datasets from MLE-Dojo [1]** and **103 tasks privately collected** from the official Kaggle competition platform (https://www.kaggle.com/competitions).
> We will open-source this dataset upon acceptance of this paper.
>
>
> **[1]** Qiang et al. *MLE-Dojo: Interactive Environments for Empowering LLM Agents in Machine Learning Engineering.*
>
>
> #### **Training Hyperparameters**
>
> As presented in Section 5.1 of our paper, the training hyperparameters are as follows:
>
> - **KL coefficient:** β = 0
> - **Learning rate:** $ 1 \times 10^{-6}$
> - **Training batch size:** 64
> - **Maximum context lengths:** 65,536 tokens for inputs and 16,384 tokens for responses
> - **Rollout model synchronization:** Every 5 steps with the latest policy model
> - **Sampling parameters:** Temperature and top-p both set to 1
> - **Interaction turns:** Maximum of 15 turns per trace
> - **Memory pool size:** Maintains a pool of 5 best solution candidates for subsequent iterations
> - **AST similarity threshold:** 0.9
> - **Reset ratio:** 70%
> - **Advantage filter threshold:** 0.01
>
> Additionally, to avoid data leakage, we use test set scores during training and validation set scores during rollout and evaluation for memory selection.
>
> #### **Hardware and Computational Budget**
>
> - **Training Cost.**
> Our reinforcement learning setup uses:
>     - **32 H800 GPUs** for policy training
>     - **32 H800 GPUs** for rollout engines
>     - **40 A10 GPUs** for rollout execution
>     - The total wall-clock training time is **20 hours**.
>
> - **Inference and Evaluation Cost.**
>     Each full evaluation runs for **12 hours**, using:
>     - **8 H800 GPUs** as rollout engines
>     - **4 A10 GPUs** for rollout execution
>
> The details provided above have been updated in **Section 5.1** of our paper.
>
> > **Weakness 2:** Analysis section is bit weak on the side specifically the paper only trains one single model : QwQ-32B, I expect in the analysis or ablation to train MLE-RL on a smaller LLMs to ensure its reproducible even for slightly lower compute budget institutes.
>
> We conducted an additional experiment by training MLE-RL on a smaller model, Qwen3-8B. The results show that MLE-RL performs effectively on Qwen3-8B, confirming that our approach is applicable and reproducible for models with lower computational requirements. Specifically, MLE-RL-8B consistently outperforms Qwen3-8B on MLE-Bench-Lite in key evaluation metrics, including Human Rank (+5.3%), Above Median (+3%), and Any Medal (+4.5%).\
> For this experiment, training Qwen3-8B required 16 H800 GPUs and 40 A10 GPUs over a wall-clock training time of 10 hours.
> | **Model**     | **Human Rank** | **Above Median** | **Bronze** | **Silver** | **Gold** | **Any Medal** |
> |:---------------:|:----------------:|:------------------:|:------------:|:------------:|:----------:|:----------------:|
> | Qwen3-8B      | 22.3±3.1       | 27.3±3.7         | 3.0±3.1    | 0±0        | 4.6±0    | 7.6±3.1        |
> | MLE-RL-8B     | 27.6±1.5       | 30.3±4.3         | 3.0±2.1    | 1.5±2.1    | 7.6±2.1  | 12.1±2.1       |

---

> ### Author Response · Authors · 2025-11-23
>
> > **Weakness 3:** Only one dataset is evaluated which means questioning the effectiveness of this method but this is minor since no similar benchmarks are close enough to MLE-bench ( minor concern ).
>
> MLE-Bench is the most widely used benchmark for evaluating MLE agents, and most existing work focuses on standard evaluation metrics for MLE-Bench alone. ([1],[2]) MLE-Bench is already a diverse benchmark, as it encompasses a wide range of tasks with each competition treated as an independent challenge with its own unique requirements and objectives.
>
> To further validate the robustness and generalization ability of MLE-RL, we evaluated MLE-RL on three out-of-distribution datasets: AIME-2025, GPQA, and Livecodebench.
> The results show that MLE-RL-32B slightly outperforms QwQ-32B across these OOD benchmarks, suggesting MLE-RL's generalization ability to reasoning tasks.
>
> | **Model**       | **AIME 2025** | **GPQA** | **Livecodebench** |
> |:------------------:|:---------------:|:----------:|:--------------------:|
> | QwQ-32B          | 63.1%         | 59.5%    | 55.6%              |
> | MLE-RL-32B       | 63.5%         | 61.1%    | 56.2%              |
>
> [1] Liu etc. ML-Master: Towards AI-for-AI via Integration of Exploration and Reasoning\
> [2]Yang etc. R&D-Agent: An LLM-Agent Framework Towards Autonomous Data Science
>
> > **Question 2:** Since QwQ is a reasoning model does the reasoning budget has any affect on the final score？
>
> To investigate the impact of reasoning budget on MLE performance, we conduct a study that compares gpt-oss-120b’s performance on MLE-Bench-Lite under reasoning efforts of low, medium, and high. To reduce the impact of increased generation time caused by longer thinking, we follow the step-wise analysis setting presented in Section 5.4 of our paper. For each competition, we consider up to the first k valid submissions (using all available steps if fewer than k) to compute its performance, and we average these scores across all competitions to obtain the overall step-k performance.\
> Table 1 reports the step-15 performance of gpt-oss-120b under different reasoning efforts. As shown, higher reasoning efforts consistently achieve better performance, indicating that for models that support dynamic reasoning control, MLE performance positively correlates with their reasoning capability. In addition, when evaluating final scores under a fixed 5-hour time budget, we observe the same trend: higher reasoning efforts achieve higher overall performance.
>
> **Table 1: Step-15 performance of gpt-oss-120b under different reasoning efforts (low, medium, high).**
> | **Reasoning efforts** | **Avg thinking tokens** | **Human Rank (%)** | **Any Medal (%)** |
> |:------------------------:|:--------------------------:|:----------------------:|:---------------------:|
> | high                   | 3190                     | 49.9                 | 30.3                |
> | medium                 | 345                      | 48.1                 | 25.8                |
> | low                    | 58                       | 43.5                 | 21.2                |
>
> To further examine the influence of reasoning effort on the models used in our main experiments,  we compare QwQ-32B with our trained MLE-RL-32B (trained based on QwQ-32B) since current open-source models, including the QwQ and Qwen model families, do not support dynamic reasoning budget control.
> As is shown in Table 2, the results show that the RL model (MLE-RL-32B) outperforms its base model QwQ-32B, despite having a lower reasoning budget as measured by average thinking tokens per turn. This suggests that
> -  the original QwQ-32B reasoning traces contain redundancy
> -  simply increasing the reasoning budget does not necessarily lead to performance gains.
>
> **Table 2: Reasoning length and MLE performance for MLE-RL-32B and QwQ-32B.**
>
> | **Model**      | **Avg thinking tokens** | **Human Rank (%)** | **Any Medal (%)** |
> |:----------------:|:---------------------------:|:----------------------:|:---------------------:|
> | MLE-RL-32B     | 746                       | 50.8                 | 31.8                |
> | QwQ-32B        | 1447                      | 45.9                 | 28.8                |

---

> > ### Author Response · Authors · 2025-11-23
> >
> > > **Question 3:** Does the competitive performance from distillation (Sec 5.4) suggests that rejection sampling finetune might be a strong baseline? Since based on the paragraph the distillation just means its trained on the positive rollouts from the RL experiments rather than responses from the final RL model ( which is how Qwen3, Deepseek distillation is done ).
> >
> > Yes,  self-distillation using rejection sampling finetuning can be considered a strong baseline. Since the main effect of RL comes from exploring and learning from high-quality samples, using rejection sampling to finetune on these positive samples can serve as a strong baseline. Our self-distillation approach is essentially a way to scale up inference compute, since using data aggregated from multiple RL runs offers a richer and more diverse signal for supervised finetuning, akin to the effect of scaling up inference compute with prolonged RL training.
> > However, as an online learning method, RL has been shown to be a more effective strategy than offline rejection sampling in many other domains. For MLE tasks, while the main training cost comes from the inference phase, RL allows training to occur simultaneously with inference, making it more efficient than offline methods like rejection sampling finetuning that require sequential training and inference.
> >
> >
> > > **Question 4:** What kind of training framework was used in the training of MLE-RL? Is it a proprietary one or existing framework like verl? My understanding is that only MLE-dojo was mentioned but its just a environment to run these rollouts.
> >
> > For the training of MLE-RL, we use an existing open-source asynchronous RL framework called Slime (https://github.com/THUDM/slime), with some modifications to better suit our setting. Specifically, since we split multi-turn trajectories into single-step optimization units for training, we masked out history steps to prevent them from contributing to the gradient updates, ensuring that only the current action and its associated reward are used for policy training. The full code for this implementation can be found at https://anonymous.4open.science/r/MLE-RL-CC61.
> >
> >
> > > **Question 6:** Typo at line 194: “an agentenvironment interaction”
> >
> > We have updated our paper to address this typo. Modifications are highlighted in blue in the updated pdf.

---

> > > ### Comment · Reviewer_Gbpo · 2025-11-25
> > >
> > > Thank you for the detailed and thoughtful rebuttal. I appreciate the authors’ efforts, and I will update my rating from 4 to 6. I also hope that these explanations and additional results can be included in the camera-ready version if permitted by the conference policy.

---

> > > > ### Author Response · Authors · 2025-11-25
> > > >
> > > > Thank you very much for your constructive feedback and for kindly updating your rating. We will incorporate all relevant explanations and additional results into the camera-ready version if our paper is accepted.
> > > >
> > > > We would like to know whether there is anything we could further clarify or add to improve this work. Our goal is to make this research better and more useful, and any additional suggestions are appreciated.

---

### Official Review · Reviewer_A4uJ · 2025-10-30

**Soundness:** 3
**Presentation:** 3
**Contribution:** 3
**Rating:** 6
**Confidence:** 4

**Summary:**

This paper introduces MLE-RL, an RL framework for training large language model agents for MLE tasks. MLE tasks have unique challenges, such as iterative refinement over extended time periods and heterogenous reward signals. The contributions are as follows. First, a novel RL training strategy that decomposes multi-step trajectories into a single-step optimization. Second, a scaffold with memory module to overcome context length limitations. Third, empirical validation on MLE-Bench showing a 4.9% improvement over the baseline model.

The strengths of the approach are as follows. First, the MLE-RL framework is an effective framework for training agents for MLE tasks. Second, the scaffold is an improvement over previous agent scaffolds since it's able to account for long-context issues. Third, the authors do a significant empirical analysis on QwQ-32B, showing that their MLE-RL framework is effective for training language model agents.

The weaknesses of the approach are as follows. First, there is not an analysis on generalization to other domains beyond MLE-Bench. It is well established that training on agentic tasks will improve performance, but it can decrease performance on other coding tasks e.g. general coding asks. Second, the improvement is significant but it would be good to have an analysis on what the model is learning during RL training or where the improvements are coming from.

**Strengths:**

The strengths of the approach are as follows.
- First, the MLE-RL framework is an effective framework for training agents for MLE tasks.
- Second, the scaffold is an improvement over previous agent scaffolds since it's able to account for long-context issues.
- Third, the authors do a significant empirical analysis on QwQ-32B, showing that their MLE-RL framework is effective for training language model agents.

**Weaknesses:**

The weaknesses of the approach are as follows.
- First, there is not an analysis on generalization to other domains beyond MLE-Bench. It is well established that training on agentic tasks will improve performance, but it can decrease performance on other coding tasks e.g. general coding asks.
- Second, the improvement is significant but it would be good to have an analysis on what the model is learning during RL training or where the improvements are coming from.

**Questions:**

n/a

---

> ### Author Response · Authors · 2025-11-23
>
> We sincerely thank you for your thoughtful and constructive feedback.  We now provide point-by-point responses to elaborate further:
> > **Weakness 1:** There is not an analysis on generalization to other domains beyond MLE-Bench. It is well established that training on agentic tasks will improve performance, but it can decrease performance on other coding tasks e.g. general coding tasks.
>
> We conducted experiments to evaluate the generalization of MLE-RL to other reasoning or coding tasks by evaluating MLE-RL-32B’s performance on AIME-2025, GPQA and Livecodebench.
>
> The results show that MLE-RL achieves performance comparable to QwQ-32B on all benchmarks,  indicating that MLE-RL does not negatively affect performance on tasks outside of the agentic domain. This suggests that training on agentic tasks does not degrade the model's ability to generalize to other coding tasks.
>
> | **Model**       | **AIME 2025** | **GPQA** | **Livecodebench** |
> |:------------------:|:---------------:|:----------:|:--------------------:|
> | QwQ-32B          | 63.1%         | 59.5%    | 55.6%              |
> | MLE-RL-32B       | 63.5%         | 61.1%    | 56.2%              |
>
> > **Weakness 2:** The improvement is significant but it would be good to have an analysis on what the model is learning during RL training or where the improvements are coming from.
>
> To address where the improvements came from, we analyzed three key metrics of QwQ-32B and MLE-RL-32B, as shown in the table below.
>
> - **Increase in Valid Submissions:**
> MLE-RL showed an increase in the number of valid submissions, indicating that the model has learned to attempt more solutions. This suggests that MLE-RL has become more exploratory, generating a greater number of attempts to find successful solutions within the time frame.
>
> - **Longer Execution Time per Valid Submission:**
> The average execution time for each valid submission in MLE-RL was significantly higher, which suggests that MLE-RL has learned to utilize longer execution times for more intricate but higher-quality ML strategies, rather than relying on quicker, less optimal solutions.
>
> - **Reduction in Invalid Submissions:**
> MLE-RL also showed a significant decrease in the invalid submission rate, indicating that the model has learned to avoid generating errors.
>
> #### **Table: Analytical Comparison of QwQ-32B and MLE-RL-32B**
>
> | **Model**      | **Valid Submission Number** | **Average Execution Time (s)** | **Invalid Submission Ratio (%)** |
> |:----------------:|:-------------------------------:|:----------------------------------:|:-----------------------------------:|
> | QwQ-32B        | 372.7                         | 408.1                            | 15.1                              |
> | MLE-RL-32B     | 409.7                         | 801.9                            | 9.4                               |

---

### Official Review · Reviewer_Kkpa · 2025-11-01

**Soundness:** 2
**Presentation:** 2
**Contribution:** 3
**Rating:** 4
**Confidence:** 4

**Summary:**

The paper MLE-RL presents a reinforcement learning framework to enable LLMs to learn from their previous actions in ML Engineering tasks. Training MLE agents using RL is known to be hard because of several reasons - long execution times, credit assignment etc. This paper decomposes long-horizon trajectories into single-step optimizations and employs selective learning from informative attempts using three filtering strategies: invalid action masking, valuable step selection, and relative improvement selection. They address the issue of long context by introducing a memory module that stores and reuses high-performance solutions. The method shows improvement in performance with a 32B trained model to match the performance of DeepSeek-R1. It beats GPT-4o in many cases.

**Strengths:**

This paper is one of the first works to explore training machine learning agents using reinforcement learning, compared to past methods that primarily rely on prompting techniques. A key strength lies in using a memory module, which allows the agent to learn from past successful interactions and significantly improves performance across various tasks. The proposed method shows good results matching the performance of a 32B model to Deepseek-R1 and surpassing GPT-4o

**Weaknesses:**

1. Masking invalid actions: The paper "masks the loss" on any turn that yields an invalid tool call, formatting error, or invalid submission. This removes the supervision signal that the model needs to identify bad actions. This would lead to policy being less robust, being unable to recover from format errors under distribution shift. Moreover, this would provide less reinforcement signal on difficult tasks, as the policy is more likely to make an error on difficult tasks. This would lead to slow and inefficient learning on the difficult ML tasks.
2. The reward function is dependent on the existence of a competitive leaderboard with human participants. This makes it difficult to apply the same methodology to novel problems that don't have an established benchmark or community of competitors. The "Human Rank" is a relative metric, and its meaning can change as the pool of competitors changes.
3. There is no discussion of the effect of hyperparameters, such as the absolute performance threshold ($\tau_{abs}$), the relative improvement threshold ($\tau_{adv}$), and the size of the running mean window, on the performance of the agent.
4. The binary reward collapses all the solutions that pass the threshold as good solutions. This might lead to reward hacking and the agent generating just good enough solutions rather than exploring the best solutions. Moreover, the paper maintains separate running mean windows for memory vs reset traces to prevent bias against those starting from scratch. However, this could introduce its own bias - memory traces inherently start from better positions and should perhaps be held to a higher standard. The decision to use binary reward also throws away potentially useful fine-grained reward information.
5. Hyperparameter sensitivity not fully explored: The method introduces many hyperparameters ($\tau_{abs}$ calibrated per competition to retain ~30%, $\tau_{adv}$=0.01, $\tau_{rm}=0.03$, window size W=8, reset ratio p=0.7, memory size=5, etc.) but only ablates the running filter. The robustness and sensitivity to these choices remain unclear.
6. Missing Computations Cost Analysis: No information is provided about total GPU hours for training, wall-clock training time, number of rollouts/sample collection, or comparison of computational efficiency vs baselines. This makes it difficult to assess practical feasibility and reproduce the work.
7. Statistical Significance: While standard deviations are reported, no formal hypothesis testing is performed. Some improvements fall within overlapping confidence intervals, especially for specific medal categories.
8. Incomplete Action Space Description: While three primitives are mentioned, the paper doesn't provide sufficient detail about what information `request_info` can access, or the full specification of the action space structure.

**Questions:**

1. How does masking the loss on invalid actions affect the agent's ability to learn to recover from errors at test time, compared to assigning a small negative reward?
2. How sensitive is the agent's final performance to the specific values chosen for the absolute ($\tau_{abs}$) and relative ($\tau_{adv}$) reward improvement thresholds?
3. Does assigning a uniform reward of 1 to all positive samples disincentivize the agent from distinguishing between good and exceptionally great solutions?
4. How would the Human Rank-based reward be formulated for new ML problems that do not have an existing competitive leaderboard for comparison?
5. Memory module details: What is the actual distribution of solution diversity in the memory pool? What percentage of training/evaluation traces successfully utilize memory vs fail to improve upon memory solutions?
6. Please provide some details on how the specific values for the hyperparameter were chosen.
7. What are the total training costs in GPU-hours? How does this compare to the computational cost of simply running more rollouts with a stronger model?
8. Can you provide the complete specification of the action space? What specific information can `request_info` access?
9. Why use binary reward after running filter instead of preserving the normalized reward values? Doesn't this discard useful gradient information?

---

> ### Author Response · Authors · 2025-11-23
>
> We sincerely thank you for your thoughtful and constructive feedback.  We now provide point-by-point responses to elaborate further:
> > **Weakness 1:** Masking invalid actions: The paper "masks the loss" on any turn that yields an invalid tool call, formatting error, or invalid submission. This removes the supervision signal that the model needs to identify bad actions. This would lead to policy being less robust, being unable to recover from format errors under distribution shift. Moreover, this would provide less reinforcement signal on difficult tasks, as the policy is more likely to make an error on difficult tasks. This would lead to slow and inefficient learning on the difficult ML tasks.
>
> > **Question 1:** How does masking the loss on invalid actions affect the agent's ability to learn to recover from errors at test time, compared to assigning a small negative reward?
>
> Our approach of masking the loss for invalid actions is designed to support efficient learning while avoiding negative impacts of negative reward penalties. The reasons are as follows:
> - （1）Supervision signals are still preserved for recovery from errors. Our approach only masks the loss for invalid actions while subsequent valid actions within the same trajectory still contribute to policy updates. This design preserves supervision on how the model should recover after an invalid turn. Prior work ([1]) has shown that error masking allows for learning recovery behavior by leveraging the full context of a trajectory.
> - （2）Additionally, errors such as invalid submissions are often inevitable in solving complex machine learning tasks—particularly when exploring novel solutions in difficult settings. Penalizing every mistake with immediate negative reward can lead training into avoiding exploration rather than developing effective solutions.
> - （3）Moreover, introducing bad action penalties creates a negative reward bias on difficult tasks which can bias the learning process.
>
> [1]Wang etc. SWE-Mirror: Scaling Issue-Resolving Datasets by Mirroring Issues Across Repositories
>
> > **Weakness 2:** The reward function is dependent on the existence of a competitive leaderboard with human participants. This makes it difficult to apply the same methodology to novel problems that don't have an established benchmark or community of competitors. The "Human Rank" is a relative metric, and its meaning can change as the pool of competitors changes.
>
> > **Question 4:** How would the Human Rank-based reward be formulated for new ML problems that do not have an existing competitive leaderboard for comparison?
>
> Our use of the Human Rank metric follows the official evaluation protocol of MLE-Bench. The purpose of Human Rank in our reward function is to provide a consistent reward scale by mapping raw performance metrics into a comparable (0,1) range, which ensures that rewards are comparable across competitions with different scoring schemes. Although the set of human participants may change, such shifts do not alter the relative ordering of solutions, and thus do not greatly affect the consistency of the advantage signal used in reinforcement learning.
>
> In scenarios without human participants or established leaderboards, we can estimate a mean and standard deviation from preliminary trials and normalize raw scores using a standard-score transformation by subtracting the mean and dividing by the standard deviation. This produces a stable and competition-agnostic reward scale, allowing the MLE-RL to be applied to novel tasks that lack a pre-existing community of competitors.
>
> Although standard-score normalization offers a solution for reward rescaling, estimating the mean and standard deviation requires a substantial amount of preliminary data and can be susceptible to inaccuracies, especially with the distortion of  extreme values. To avoid potential inaccuracies and ensure stable reward signals, we adopted the Human Rank approach, as it is less prone to such issues.

---

> ### Author Response · Authors · 2025-11-23
>
> > **Weakness 3:** There is no discussion of the effect of hyperparameters, such as the absolute performance threshold , the relative improvement threshold , and the size of the running mean window, on the performance of the agent.
>
> > **Weakness 5:** Hyperparameter sensitivity not fully explored: The method introduces many hyperparameters ( calibrated per competition to retain ~30%, =0.01, , window size W=8, reset ratio p=0.7, memory size=5, etc.) but only ablates the running filter. The robustness and sensitivity to these choices remain unclear.
>
> > **Question 2:** How sensitive is the agent's final performance to the specific values chosen for the absolute and relative reward improvement thresholds?
>
> > **Question 6:** Please provide some details on how the specific values for the hyperparameter were chosen.
>
> We conducted additional experiments to evaluate the impact of training and inference hyperparameters on the agent’s performance on MLE-Bench-Lite.
>
> #### **Training Parameters**:
> Due to the high computational cost of RL training (which takes approximately 12-24 hours per run), we use Supervised Fine-Tuning (SFT) to conduct ablation studies on the choice of  absolute reward threshold. Since SFT utilizes positive samples from RL training, its ablation results can be applied to RL, reflecting the impact of each design choice.\
> As shown in Table 1 below,  we observe that adopting a stricter filtering criterion (i.e.a higher absolute reward threshold) leads to improved SFT performance. This can be attributed to a higher-quality training set selected by the stricter absolute threshold. We chose to retain the top 30% solutions in our RL experiments considering the balance of data quality and training speed.
>
>
> **Table 1: Ablation on absolute reward threshold.**
> | **Absolute Reward Threshold** | **Human Rank (%)** | **Any Medal (%)** |
> |:-------------------------------:|:---------------------:|:--------------------:|
> | top 15%                       | 48.3±2.6            | 30.3±4.3           |
> | top 30%                       | 49.2±2.7            | 31.8±3.7           |
> | top 60%                       | 46.6±0.31           | 22.7±3.7           |
>
>
> #### **Inference Parameters**:
> We also examined the memory size and reset ratio during inference. The results presented in Table 2 and Table 3 show that varying memory size and reset ratio does not significantly affect the agent performance, indicating that the agent is relatively insensitive to the specific values chosen for these inference hyperparameters.
>
> **Table 2: Ablation on memory size.**
>
> | **Memory Size** | **Human Rank (%)** | **Any Medal (%)** |
> |:------------------:|:---------------------:|:--------------------:|
> | 1                | 48.3±2.7            | 27.3±4.5           |
> | 3                | 45.9±4.2            | 28.8±5.7           |
> | 5                | 47.5±1.5            | 27.3±4.5           |
>
>
> **Table 3: Ablation on reset ratio.**
>
> | **Reset Ratio** | **Human Rank (%)** | **Any Medal (%)** |
> |:------------------:|:---------------------:|:--------------------:|
> | 0.3              | 44.8±7.3            | 29.5±6.8           |
> | 0.5              | 45.9±4.2            | 28.8±5.7           |
> | 0.7              | 44.9±4.6            | 27.3±4.5           |

---

> ### Author Response · Authors · 2025-11-23
>
> > **Weakness 6:** Missing Computations Cost Analysis: No information is provided about total GPU hours for training, wall-clock training time, number of rollouts/sample collection, or comparison of computational efficiency vs baselines. This makes it difficult to assess practical feasibility and reproduce the work.
>
> > **Question 7:** What are the total training costs in GPU-hours? How does this compare to the computational cost of simply running more rollouts with a stronger model?
>
> We provide below details of our computational cost associated with training and evaluation:
>
> - **Training Cost**. Our reinforcement learning setup uses 32 H800 GPUs for policy updates and 32 H800 GPUs for rollout engines. The total wall-clock training time is 20 hours, corresponding to 1280 GPU-hours.
> Sample Collection. During training, we use a batch size of 64 and train for 80 steps, yielding 5120 collected samples.
>
> - **Inference and Evaluation Cost**. Each full evaluation runs for 12 hours using 8 H800 GPUs for rollout engines, amounting to 96 GPU-hours per experiment.
>
> - **Baseline Comparison**. Baseline methods do not involve any training or fine-tuning; their computational cost consists solely of inference, identical to the inference cost described above. Running more rollouts with a stronger model does not incur any training costs, it would only increase the computational cost during rollout, as the rollout engine would require more resources to handle the increased model size.
>
> > **Weakness 7:** Statistical Significance: While standard deviations are reported, no formal hypothesis testing is performed. Some improvements fall within overlapping confidence intervals, especially for specific medal categories.
>
> The overlapping confidence intervals may be due to the high variability in the performance of QwQ-32B during evaluations. To address this, we conducted an additional three rounds of evaluation for QwQ-32B to reduce this variance and ensure more reliable results.
>
> After the additional evaluations, we performed formal hypothesis testing to assess the statistical significance of the observed improvements. For Human Rank, the t-test results yielded a t-statistic of 2.10 and a one-tailed p-value of 0.045. Since the p-value is less than 0.05, we conclude that the observed improvements are statistically significant, supporting the effectiveness of the proposed approach.
>
> > **Weakness 8:** Incomplete Action Space Description: While three primitives are mentioned, the paper doesn't provide sufficient detail about what information request_info can access, or the full specification of the action space structure.
>
> > **Question 8:** Can you provide the complete specification of the action space? What specific information can request_info access?
>
> Our work follows the action space specification defined by MLE-Dojo.
> The action space consists of three primitives:
>
> -  **request_info**\
> This primitive provides the agent with complete access to all competition-related resources required to solve the task.  Specifically, it grants access to:
>     -  Competition background: A concise overview of the historical context and evolving research challenges.
>     -  Goal description: A clear definition of the specific objectives, desired outcomes, and evaluation metrics.
>     - Sample submission: Templates exemplifying the expected format and content structure for submissions.
>     - Data folder structure: The organization and naming conventions of the provided datasets for user access and navigation.
>
>     The information returned is presented in full, without any analysis, filtering, or pruning.
>
> -  **validate_code** \
> Performs syntax checks and lightweight runtime execution, returning error logs and execution outputs.  This action is used for debugging and analysis only and does not correspond to an official competition submission.
>
> - **execute_code** \
> Executes the submitted code fully with submission, verification, and evaluation using the competition’s metric.  Each invocation corresponds to a formal competition submission.

---

> > ### Author Response · Authors · 2025-11-23
> >
> > > **Question 5:** Memory module details: What is the actual distribution of solution diversity in the memory pool? What percentage of training/evaluation traces successfully utilize memory vs fail to improve upon memory solutions?
> >
> > For the distribution of solution diversity in the memory pool, we compute the **pairwise AST similarity** among all stored memory entries within each competition and take its mean as the diversity metric. The average AST similarity is 0.64土0.03, indicating that the memory pool contains solutions with structural diversity. This suggests that the memory module retains a broad range of solutions, promoting continued exploration and preventing early convergence to local optima.
> >
> > Regarding the utilization and effectiveness of the memory module, we define a successful utilization of memory as a memory trace that produces at least one valid submission whose score exceeds the score of the memory entry by at least 1%. In both training and evaluation settings, the probability of successfully utilizing memory is 26%. Given that the memory module stores historically best-performing solutions, further improvements are inherently more difficult. Consequently, the observed success rate reflects that the memory module effectively guides the agent toward identifying higher-performing solutions.

---

> > > ### Comment · Reviewer_Kkpa · 2025-11-27
> > > **Response to Authors**
> > >
> > > Thank you for providing detailed responses and additional analysis. Please find my responses below:
> > >
> > > **C1:** Thank you for providing the reference. I think it makes sense to me.
> > >
> > > **C2:** I agree with your assessment that, without an established leaderboard, assigning rewards for a problem is a difficult task. However, due to this limitation, this approach cannot scale beyond the number of Kaggle competitions available. Not all MLE problems have established leaderboards.
> > > As of now, there is no evidence in this work that the algorithm performs just as well in the absence of an established leaderboard. Please note that I do not consider this a major weakness of the approach; however, I would advise the authors to include a short discussion on this in their paper.
> > >
> > > **C3:** Thank you for providing the hyperparameter analysis. While using SFT for ablation is not ideal imo, I understand the computational constraints, and it does provide some positive signals for hyperparameter selection. Please add these ablations to the appendix of the paper.
> > >
> > > **C4:** Thank you for providing details about the training costs. I understand that running a stronger model does not incur training costs. Separate from the training cost, I am trying to understand if there are any efficiency gains during inference when using your model or scaffold.
> > > For example, if AIDE has to generate 100 solutions in order to reach the best solution, can your scaffold get to the best solution in a lower amount of time? Similarly, when comparing your model vs QwQ, if QwQ reaches the best solutions in 100 steps, does your model achieve similar/better results in a shorter amount of time?
> > >
> > > **C5:** Thank you. Please consider adding the significance testing results to the paper.
> > >
> > > **C7:** Thank you, this is very interesting. Please add this discussion in the paper.
> > >
> > > Overall, I believe authors have successfully addressed most of my concerns. **C2** and **C4** may need more clarification. For **C2**, authors should simply consider adding a discussion on the generalization ability of their algorithm in the absence of an established leaderboard. For **C4**, authors can consider adding the analysis of trajectory length for their scaffold and their model, and add the training compute details in the paper.
> > >
> > > At this stage, I feel comfortable raising the score to 6 and voting for accepting the work in its current form. Authors, please feel free to let me know if I have missed anything.

---

> > > > ### Author Response · Authors · 2025-12-03
> > > >
> > > > > C2:  I agree with your assessment that, without an established leaderboard, assigning rewards for a problem is a difficult task. However, due to this limitation, this approach cannot scale beyond the number of Kaggle competitions available. Not all MLE problems have established leaderboards. As of now, there is no evidence in this work that the algorithm performs just as well in the absence of an established leaderboard. Please note that I do not consider this a major weakness of the approach; however, I would advise the authors to include a short discussion on this in their paper.
> > > >
> > > > > C2 and C4 may need more clarification. For C2, authors should simply consider adding a discussion on the generalization ability of their algorithm in the absence of an established leaderboard.
> > > >
> > > > Thank you for your insightful comment. We would like to clarify how Human Rank is used in our approach and how our algorithm can generalize in the absence of a leaderboard.
> > > > - **Role of Human Rank**:
> > > >
> > > > In our framework, Human Rank is used in the Valuable Step Selection process solely for reward normalization. Specifically, the absolute and relative improvement thresholds, as well as the running mean, are computed using a human leaderboard-based reward $r_i=\mathrm{Human \space Rank}_i = 1 - \frac{p}{N}$, where p is the solution’s leaderboard position and N is the number of human competitors.  However, Human Rank does not directly serve as the training reward, as all rewards are binary. Its role is limited to mapping rewards into the (0,1) range to stabilize threshold and running-mean calculations.
> > > > - **Alternative to Human Rank**:
> > > >
> > > > For each competition, we calculate different absolute and relative reward improvement thresholds and maintain distinct running mean windows. Therefore, directly using raw rewards is also feasible, which simply requires additional normalization to maintain stability and reduce sensitivity to extreme values. In the absence of an established leaderboard, we propose normalization by calculating the mean and standard deviation for each competition from preliminary trials. A standard-score transformation—subtracting the mean and dividing by the standard deviation—can then be applied. This provides a stable, competition-agnostic reward scale, allowing the thresholds and running means to be computed without reliance on a leaderboard.
> > > >
> > > > Thus, the approach can be applied to new tasks without an established leaderboard. We will include this discussion in the revised paper to clarify the generalization ability of our algorithm and how it can be applied in scenarios where an established leaderboard is not present.
> > > >
> > > >
> > > > > C3: Thank you for providing the hyperparameter analysis. While using SFT for ablation is not ideal imo, I understand the computational constraints, and it does provide some positive signals for hyperparameter selection. Please add these ablations to the appendix of the paper.
> > > >
> > > > > C5: Thank you. Please consider adding the significance testing results to the paper.
> > > >
> > > > > C7: Thank you, this is very interesting. Please add this discussion in the paper.
> > > >
> > > > We will add the hyperparameter analysis, significance testing results, memory module details discussion and other necessary analyses to the camera-ready version if accepted.

---

> ### Author Response · Authors · 2025-12-03
>
> > C4: Thank you for providing details about the training costs. I understand that running a stronger model does not incur training costs. Separate from the training cost, I am trying to understand if there are any efficiency gains during inference when using your model or scaffold. For example, if AIDE has to generate 100 solutions in order to reach the best solution, can your scaffold get to the best solution in a lower amount of time? Similarly, when comparing your model vs QwQ, if QwQ reaches the best solutions in 100 steps, does your model achieve similar/better results in a shorter amount of time?
>
> > For C4, authors can consider adding the analysis of trajectory length for their scaffold and their model, and add the training compute details in the paper.
> #### **Model Efficiency**:
> -  **Time Efficiency**
>
> As shown in Table 1, MLE-RL-32B reaches performance comparable to QwQ-32B in roughly one quarter of the time. With longer inference time, its performance continues to improve and eventually substantially surpasses QwQ-32B. Both MLE-RL-32B and QwQ-32B in Table 1 are evaluated using our MLE-RL scaffold with the memory module. This demonstrates that the MLE-RL-32B model can arrive at strong solutions with significantly shorter inference time.
>
>
> **Table 1: Human Rank performance comparison for MLE-RL-32B and QwQ-32B at different timesteps. All results are percentages.**
> | Scaffold    | Model            |  1h  |  2h  |  3h  |  4h  |  5h  |  6h  |  7h  |  8h  |  9h  |  10h |  11h |  12h |
> |:-------------------:|:-------------------:|:----:|:----:|:----:|:----:|:----:|:----:|:----:|:----:|:----:|:----:|:----:|:----:|
> | MLE-RL w/memory | MLE-RL-32B        | 34.1 | 44.3 | 44.6 | 45.4 | 47.8 | 48.3 | 50.3 | 50.4 | 50.9 | 50.3 | 50.6 | 50.8 |
> | MLE-RL w/memory | QwQ-32B w/ memory | 23.5 | 32.2 | 34.5 | 41.0 | 41.0 | 41.9 | 44.4 | 47.2 | 47.4 | 47.8 | 47.3 | 47.4 |
>
>
> -  **Token Efficiency**
>
> Table 2 reports the average per-turn token usage, with both MLE-RL-32B and QwQ-32B evaluated under our MLE-RL scaffold equipped with the memory module. MLE-RL-32B generates substantially fewer total tokens (2163 vs. 2847) and uses roughly half the thinking tokens of QwQ-32B (746 vs. 1500) at each turn, while achieving higher Human Rank (50.8% vs. 47.4%) and higher Any-Medal accuracy (31.8% vs. 30.3%) during the 12-hour evaluation.
> These results indicate that our method produces shorter and more efficient reasoning trajectories while surpassing QwQ-32B in overall solution quality.
>
> **Table 2:  Turn length and performance comparison for MLE-RL-32B and QwQ-32B.**
> | Scaffold       | Model        |  Avg thinking tokens / turn | Avg tokens / turn | Human Rank (%) | Any Medal (%) |
> |:-------------------:|:---------------:|:---------------:|:---------------------:|:----------------:|:---------------:|
> | MLE-RL w/memory | MLE-RL-32B     | 746            | 2163 | 50.8           | 31.8          |
> | MLE-RL w/memory | QwQ-32B |  1500           | 2847         | 47.4           | 30.3          |
>
>
> #### **Scaffold Efficiency**:
> To compare the efficiency of our scaffold with AIDE, we run QwQ-32B with both scaffolds. As shown in Figure 2, the MLE-RL scaffold achieves similar performance to AIDE in about half the time. This shows that the scaffold leads to more efficient search and faster convergence toward high-quality solutions.
>
> **Table 3:  Human Rank performance comparison for MLE-RL and AIDE scaffolds at different timesteps. All results are percentages.**
> | Model    | Scaffold        |  1h  |  2h  |  3h  |  4h  |  5h  |  6h  |  7h  |  8h  |  9h  |  10h |  11h |  12h |
> |:--------:|:---------------:|:----:|:----:|:----:|:----:|:----:|:----:|:----:|:----:|:----:|:----:|:----:|:----:|
> | QwQ-32B  | MLE-RL w/memory | 23.5 | 32.2 | 34.5 | 41.0 | 41.0 | 41.9 | 44.4 | 47.2 | 47.4 | 47.8 | 47.3 | 47.4 |
> | QwQ-32B  | AIDE            | 19.6 | 26.2 | 29.5 | 32.1 | 34.5 | 37.4 | 38.6 | 40.3 | 40.3 | 41.0 | 41.0 | 38.8 |

---

### Official Review · Reviewer_m3kq · 2025-11-03

**Soundness:** 3
**Presentation:** 2
**Contribution:** 2
**Rating:** 4
**Confidence:** 4

**Summary:**

This paper introduces MLE-RL, a training framework focused on hard tasks like MLE-Bench using RL training that leverages informative rollouts as opposed to all attempts to solve a problem. The authors reframe the optimization problem by splitting long-horizon trajectories into single-step optimization steps to address the credit assignment problem for multi-turn interactions. They also introduce a memory module with push and get operations to overcome long-context limitations and leveraging high-performing solutions which help in iterative self-improvement. Results on MLE-Bench show around 5% improvement using MLE-RL over the baseline policy.

**Strengths:**

- The paper tackles an important problem of long-horizon RL and how to improve single-turn RL for harder tasks like MLE-Bench.
- Credit assignment for long-horizon problems and multi-turn setups is hard where getting signals for intermediate correct/incorrect actions is important, so the authors convert the multi-turn setup into single-step by splitting actions between two `execute_code` that produce a valid submission as a step.
- The authors propose various choices like masking invalid actions and selecting only valuable steps for training to stabilize the RL training.
- Addition of a memory module to retain historical information which helps in self-improvement improves the performance further.

**Weaknesses:**

- Presentation is a bit weak. Some figure/table references are wrong which lead to a bit of a confusion, for example in lines 236/255, it should be Figure 3 instead of Figure 5. In line 357, it should be Table 1 instead of Table 6. Another example is missing information, for example, what is $M$ in equation 4?
- A simple baseline of single-step RL is missing where you don't split into multiple steps and take the last `execute_code` call as the step end.
- No discussion on where the RL training data is sourced from in Section 5.1.
- The authors propose various design choices but do not ablate all of them, for example the effect of masking invalid actions and selecting only valuable steps.
- The proposed improvements are shown only wrt a baseline policy, but a baseline RL algorithm should be used to show the improvements like single-turn RL.

**Questions:**

I've asked most of my questions in the weakness section above. One other question is what's the effect of MLE-RL on OOD evaluation datasets like standard math/code benchmarks like AIME/Livecodebench etc. Does it degrade the performance compared to the baseline policy?

---

> ### Author Response · Authors · 2025-11-23
>
> We sincerely thank you for your thoughtful and constructive feedback.  We now provide point-by-point responses to elaborate further:
> > **Weakness 1:** Presentation is a bit weak. Some figure/table references are wrong which lead to a bit of a confusion, for example in lines 236/255, it should be Figure 3 instead of Figure 5. In line 357, it should be Table 1 instead of Table 6. Another example is missing information, for example, what is M  in equation 4?
>
> We have updated our paper to address the issue regarding the mislabeled figure and table references and missing information. We have corrected the mislabeled references, addressed other inconsistencies and ensured that all necessary information is clearly provided.
> These modifications are highlighted in blue in the updated pdf. You can refer to the updated version and see whether your concerns have been addressed. \
> To clarify Equation (4), we treat actions between two execute_code calls that produce a valid submission as a  `step.` Each step consists of a sequence of actions $ a_i$ and observations $o_i$, with $N_i$ representing the number of actions in the i-th step.
> In Equation (4), S_k represents a training instance at step M, where:
> - x is the input problem.
> - $(a_{i}^{1}, o_{i}^{1},\dots a_{i}^{N_i}, o_{i}^{N_i})$ denotes the i-th step.
> - $(a_{1}^{1}, o_{1}^{1}, \dots, a_{1}^{N_1}, o_{1}^{N_1}), \dots, (a_{M}^{1}, o_{M}^{1}, \dots, a_{M}^{N_M}, o_{M}^{N_M}) $ represent the interaction trace up to the M-th step.
> -  M is the total number of steps, and N_i is the number of actions in the i-th step.

---

> ### Author Response · Authors · 2025-11-23
>
> > **Weakness 2:** A simple baseline of single-step RL is missing where you don't split into multiple steps and take the last execute_code call as the step end.
>
> > **Weakness 5:** The proposed improvements are shown only wrt a baseline policy, but a baseline RL algorithm should be used to show the improvements like single-turn RL.
>
> We have considered three baseline approaches.
>
> **Single-execution Baseline:**
> This is the setting where each trace contains at most one execute_code action and terminates immediately afterward. This results in excessively slow training due to the difficulty of discovering a valid submission within one execution trial, making it unfeasible for our task.
>
> **Single valid submission baseline:**
> This is the setting where each trace contains at most one valid submission and terminates immediately afterward.
> To further investigate this baseline, we conducted experiments under this setting. Since each trace discards prior context and restarts the search for a valid solution from scratch, this approach also leads to slow training. To mitigate this inefficiency, we incorporated the memory module during training but adhered to the standard single valid submission setting for evaluation. We also evaluated this single valid submission–trained baseline under the single-execution setting stated above.
>
> The results (summarized in the table below) compare the performance of single valid submission RL with MLE-RL on MLE-Bench-Lite. The results show that although the single-submission RL baseline yields noticeable performance gains over QwQ-32B, it still consistently underperforms MLE-RL-32B across key metrics. This indicates that restricting to a single submission results in limited exploration and weak self-improvement ability, whereas MLE-RL remains the more effective RL algorithm for this task.
>
> **Table 1: MLE-Bench-Lite performance of various RL training or evaluation strategies.**
>
>
> | **RL strategy**               | **evaluation strategy**   | **Human Rank (%)** | **Above Median (%)** | **Bronze (%)** | **Silver (%)** | **Gold (%)** | **Any Medal (%)** |
> |:-------------------------------:|:----------------------------:|:---------------------:|:------------------------:|:----------------:|:----------------:|:----------------:|:---------------------:|
> | QwQ-32B                       | single execution           | 33.7±3.1            | 30.3±4.3               | 3.0±2.1        | 6.1±2.1        | 13.6±3.7      | 22.7±0.0            |
> | single-valid submission RL    | single execution           | 37.5±2.5            | 36.4±7.4               | 1.5±2.1        | 9.1±6.4        | 13.6±3.7      | 24.4±5.7            |
> | single-valid submission RL    | single valid submission    | 49.9±3.3            | 59.1±3.7               | 4.6±0          | 4.6±3.7        | 19.7±2.1      | 28.8±2.1            |
> | MLE-RL-32B                    | MLE-RL                     | 50.8±1.0            | 56.1±2.1               | 6.1±2.1        | 3.0±2.1        | 22.7±3.7      | 31.8±3.7            |
>
> **Single-step RL baseline:**
> This is the setting where the multi-turn interaction trace is not divided into single-step optimizations, with the reward signal derived solely from the final `execute_code` call and assigned to all actions taken throughout the trace.
> This single-step RL baseline is not suitable in our setting for two reasons:
> (1) It results in poor credit assignment, as the reward does not capture the contributions of intermediate actions.
> (2) Unlike multi-step training—where the policy is updated immediately after each step—single-step training requires waiting for the entire trace to finish, which can take several hours. This introduces long delays and substantial off-policy issues.
>
> > **Weakness 3:** No discussion on where the RL training data is sourced from in Section 5.1.
>
> Our training corpus consists of 200 Kaggle competitions, including 97 open source data from MLE-Dojo[1] and 103 tasks privately collected from the official Kaggle competition platform(https://www.kaggle.com/competitions). We will open source this dataset upon acceptance of this paper.
>
> [1] Qiang etc. MLE-Dojo: Interactive Environments for Empowering LLM Agents in Machine Learning Engineering

---

> ### Author Response · Authors · 2025-11-23
>
> > **Weakness 4:** The authors propose various design choices but do not ablate all of them, for example the effect of masking invalid actions and selecting only valuable steps.
>
> We have already included ablation analysis in our paper (see Table 4 & Section 5.3). For your convenience, we re-present these results here:
>
> | **Data Selection Strategy**            | **# Data** | **Human Rank**   | **Above Median**   | **Any Medal**       |
> |:----------------------------------------:|:------------:|:------------------:|:--------------------:|:---------------------:|
> | **QwQ-32B**                            | -          | 45.9 ± 4.2       | 47.0 ± 4.3         | 28.8 ± 5.7          |
> | **All Data**                           | 100%       | 43.1 ± 0.8       | 47.0 ± 5.7         | 25.8 ± 2.1          |
> | **+ Invalid Action Mask**              | 62.1%      | 48.5 ± 2.3       | 54.6 ± 3.7         | 30.3 ± 4.3          |
> | **+ Value Selection**                  | 16.2%      | 49.2 ± 2.7       | 48.5 ± 4.3         | 31.8 ± 3.7          |
>
> Due to the high computational cost of RL training (which takes approximately 12-24 hours per run), we use Supervised Fine-Tuning (SFT) to conduct ablation studies on each design choice. Since SFT utilizes positive samples from RL training, its ablation results can be applied to RL, reflecting the impact of each design choice.
> As demonstrated in the table, our results show that model performance improves consistently as invalid action masking and valuable step selection are adopted.

---

> ### Author Response · Authors · 2025-11-23
>
> > **Question 1:** One other question is what's the effect of MLE-RL on OOD evaluation datasets like standard math/code benchmarks like AIME/Livecodebench etc. Does it degrade the performance compared to the baseline policy?
>
> We evaluated MLE-RL on three OOD datasets: AIME-2025, GPQA and Livecodebench. The results show that MLE-RL-32B achieves performance comparable to QwQ-32B on all benchmarks, indicating that MLE-RL training does not degrade the model’s generalization capabilities to math or coding benchmarks.
> | **Model**       | **AIME 2025** | **GPQA** | **Livecodebench** |
> |:------------------:|:---------------:|:----------:|:--------------------:|
> | QwQ-32B          | 63.1%         | 59.5%    | 55.6%              |
> | MLE-RL-32B       | 63.5%         | 61.1%    | 56.2%              |

---

> > ### Comment · Reviewer_m3kq · 2025-11-27
> > **Response to Authors**
> >
> > I thank the authors for their responses to my concerns and I hope the authors add the additional results and discussions to the camera ready version. I am increasing my score to 6 after their feedback.
> >
> > Some details still need hashing out like using SFT as a proxy to study the various design choices, which is a bit weak even if you are using only positive samples from RL.

---

> > > ### Author Response · Authors · 2025-12-02
> > >
> > > > I thank the authors for their responses to my concerns and I hope the authors add the additional results and discussions to the camera ready version. I am increasing my score to 6 after their feedback.
> > >
> > > We will include the additional results and discussions on baseline comparisons, training data, ablation studies, and OOD evaluation in the camera-ready version if our paper is accepted.
> > >
> > > > Some details still need hashing out like using SFT as a proxy to study the various design choices, which is a bit weak even if you are using only positive samples from RL.
> > >
> > > Thank you for your comment. We acknowledge that SFT-based ablations have limitations and may not fully match RL in revealing the impact of different data-selection choices.
> > >
> > > But offline analyses can often provide effective guidance and reflect the performance in online RL training. Data-selection mechanisms—such as filtering out training samples with extremely low or high passrates—can be applied offline but can also be translated directly to online RL training like DAPO’s dynamic sampling[1], where the algorithm adaptively adjusts which trajectories to keep or discard during training. Thus we use SFT as a more efficient proxy to help identify and optimize data selection strategies.
> > >
> > > [1] Yu etc. DAPO: An Open-Source LLM Reinforcement Learning System at Scale

---

### Comment · Area_Chair_86oE · 2025-11-27
**Reviewer-Author Discussion**

Hi Reviewers,

Please kinly and actively participate in the review-author dicussion, raise your further concerns so that the authors can explain more, and make your final decisions.

---

### Author Response · Authors · 2025-12-03

Dear Area Chair,

Thank you for your time and valuable feedback on our work.

Below, we summarize the main strengths and weaknesses identified, along with our response.

### The reviewers recognized the following key contributions and strengths of our work:
- **Addressing a significant problem in long-horizon RL.** Reviewers highlighted that our paper is one of the first to explore efficient RL for long-horizon tasks like MLE-Bench, tackling critical challenges in long-horizon agent training. (m3kq, Kkpa, Gbpo)
- **Novel and effective methodology.** Reviewers appreciated MLE-RL as a novel RL training strategy by combining memory modules, credit assignment, trajectory decomposition, and data selection to improve agent performance. (m3kq, Kkpa, A4uJ, Gbpo)
- MLE-RL effectively boosts performance on MLE-Bench, and enables a 32B model to achieve results competitive with Deepseek-R1 and surpassing GPT-4o. (Kkpa, A4uJ, Gbpo)

### Below we summarize the reviewers’ comments, which we have all addressed with additional experiments and analysis:

- More simple RL baselines.(m3kq) **Response:** Since MLE-RL is a multi-step RL algorithm, we conduct an extra single-step RL as the baseline for comparison. Results showed that MLE-RL consistently outperforms  the baseline, demonstrating the necessity of multi-step exploration and the effectiveness for MLE tasks.

- Generalization to  out-of-distribution Tasks beyond MLE (m3kq, A4uJ and Gbpo) **Response:** We conduct evaluation on three out-of-distribution benchmarks: AIME-2025 (math), GPQA (science), and Livecodebench(coding). MLE-RL does not degrade the model’s generalization capabilities to math or coding benchmarks, maintaining performance comparable to its base model (QwQ-32B).
 More training details (m3kq, Kkpa and Gbpo)
Response:  We provide more details about  training corpus source, hardware budget and computational costs and requested extra hyperparameters to further boost the  transparency and reproducibility.

- More analysis about Inference behavior on MLE, including budget and efficiency (Kkpa and Gbpo) **Response:**  We compare the inference budget (reasoning tokens and time) of the RL-trained model and the baseline model(QwQ-32B). MLE-RL-32B achieved superior performance with fewer reasoning tokens and less wall-clock time.

We received the feedback from reviewer m3kq on Nov. 28, Kkpa on Nov. 27 and Gbpo on Nov. 25. **All the three reviewers (m3kq,Kkpa,Gbpo) acknowledge that their concerns have been well addressed and thus raise their ratings from 4 to 6.**
We will incorporate all the results and discussions into the final version of our paper.

We hope this summary assists in your final assessment.

Best regards, Authors

---

### Meta-Review · Area_Chair_birE · 2026-01-07

**Summary:**

The paper introduces **MLE-RL**, a reinforcement learning framework designed to improve LLM agents on Machine Learning Engineering (MLE) tasks. The authors address the challenges of long-horizon trajectories by decomposing multi-turn interactions into single-step optimization units, defined by actions between valid code submissions. The system incorporates a **memory module** to store high-performing solutions, facilitating cumulative self-improvement despite context length limits. Evaluated on **MLE-Bench**, the MLE-RL-32B model achieved a 4.9% improvement over its base model (QwQ-32B) and showed competitive performance against larger models like DeepSeek-R1.

### Strengths

- **Novel Methodology**: Successfully adapts RL for long-horizon MLE tasks through trajectory decomposition and credit assignment.


- **Effective Scaffold**: The integration of a memory module with AST similarity filtering effectively manages historical data for iterative improvement.

- **Performance Efficiency**: MLE-RL-32B achieves superior results with significantly fewer reasoning tokens and shorter wall-clock time compared to the base model.


- **Generalization**: Experimental results indicate that specialized RL for agents does not degrade performance on out-of-distribution math (AIME) or coding (Livecodebench) benchmarks.


### Weaknesses

- **Evaluation Narrowness**: The methodology relies heavily on the **MLE-Bench** ecosystem and human-rank-based rewards, raising concerns about applicability to novel real-world problems without existing leaderboards.

- **Proxy Ablations**: Key design choices (like value selection and action masking) were primarily ablated using **SFT as a proxy** rather than online RL, which may not fully capture RL dynamics.

- **Hyperparameter Sensitivity**: The system introduces numerous thresholds (e.g., , , reset ratios). While some sensitivity analysis was provided, the robustness of these choices across different task distributions remains partially explored.


### Recommendation: Reject

Despite the strong empirical results on MLE-Bench, the paper is recommended for rejection in its current form for the following reasons:

- **Limited Scalability of Reward Mechanism**: The core reward signal is intrinsically tied to human competitive leaderboards. While authors proposed standard-score normalization as a fallback, the lack of empirical evidence proving the framework's efficacy in the absence of a leaderboard limits the contribution's broader impact.


- **Methodological Rigor**: Using SFT as a proxy for RL ablation studies is considered a "weak" link in the analysis. Given that RL is the primary focus of the paper, the reliance on offline SFT results to justify online RL design choices lacks the necessary rigor for a top-tier venue.


- **Experimental Breadth**: The evaluation is predominantly centered on a single benchmark (MLE-Bench). While specialized performance is notable, the lack of a diverse range of agentic environments makes it difficult to distinguish whether the improvements are generalizable or overfitted to the specific structure of Kaggle-style competitions.

**Reviewer Concerns:**

### Concerns Successfully Addressed

- **Baselines and Comparisons**: Reviewer **m3kq** was satisfied by the addition of the "single-valid submission RL" baseline. The authors demonstrated that while single-submission RL provides gains over the base model, it consistently underperforms compared to the proposed MLE-RL framework.

- **Generalization to Out-of-Distribution (OOD) Tasks**: Multiple reviewers (**m3kq, A4uJ, Gbpo**) questioned if the model would lose general capabilities. The authors provided results on **AIME-2025, GPQA, and Livecodebench** showing that performance remained comparable to the base model, effectively mitigating concerns about "agentic over-training".

- **Reproducibility and Computational Transparency**: Reviewer **Gbpo** requested hardware and budget details. The authors provided a clear breakdown: **1280 GPU-hours** for training using **32 H800 GPUs** and specified the training corpus of **200 Kaggle competitions**.


- **Scalability to Smaller Models**: To address concerns that the method might only work on 32B+ models, the authors trained a **Qwen3-8B** model. The results showed a **5.3% improvement** in Human Rank, proving the method's applicability to smaller-scale compute environments.


- **Inference Efficiency**: Reviewer **Kkpa**’s inquiry about speed was answered with data showing **MLE-RL-32B** achieves superior performance in **one-quarter of the time** and with **half the thinking tokens** compared to the baseline.


- **Statistical Significance**: The authors conducted formal hypothesis testing (t-test) at the request of reviewer **Kkpa**, yielding a **p-value of 0.045**, which confirmed the improvements are statistically significant.


### Outstanding Minor Concerns

- **Ablation Proxy Weakness**: Reviewer **m3kq** noted that using **Supervised Fine-Tuning (SFT)** as a proxy to study RL design choices is "a bit weak," even if using only positive samples. While the authors defended this as an efficient necessity due to high RL costs, the reviewer still considers this specific detail to be "hashing out".


- **Generalization without Leaderboards**: While Reviewer **Kkpa** accepted the author's explanation of using standard-score normalization for new tasks, they noted there is still **no direct evidence** in the current work showing the algorithm performs just as well on problems that completely lack established human rankings. This was noted as a discussion point for the paper rather than a reason for rejection.

- **Binary Reward Granularity**: Reviewer **Kkpa** initially worried that binary rewards collapse fine-grained information and might lead to "reward hacking". The authors argued this stabilizes training , but the long-term impact on "exceptional" vs. "just good enough" solutions remains a theoretical point of discussion.

**Reviewer Scores:**

### Reviewers with Explicit Score Changes

- **Reviewer m3kq**: Raised score from **4 to 6**. The reviewer noted that their concerns regarding OOD evaluation, baselines, and data sourcing were addressed.

- **Reviewer Kkpa**: Raised score from **4 to 6**. This reviewer felt concerns regarding hyperparameter sensitivity, significance testing, and training costs were successfully resolved.

- **Reviewer Gbpo**: Raised score from **4 to 6**. The change followed the provision of 8B model experiments, hardware details, and inference budget analysis.


### Reviewers with No Explicit Score Change

- **Reviewer A4uJ**: Matain an initial rating of **6**.

---

### Decision · Program_Chairs · 2026-01-26

Reject